# Augmented Radiance Field: A General Framework for Enhanced Gaussian Splatting

**Yixin Yang**[1]     **Bojian Wu**[2]     **Yang Zhou**[1]*     **Hui Huang**[1]
[1]VCC, CSSE, Shenzhen University     [2]Tencent Games

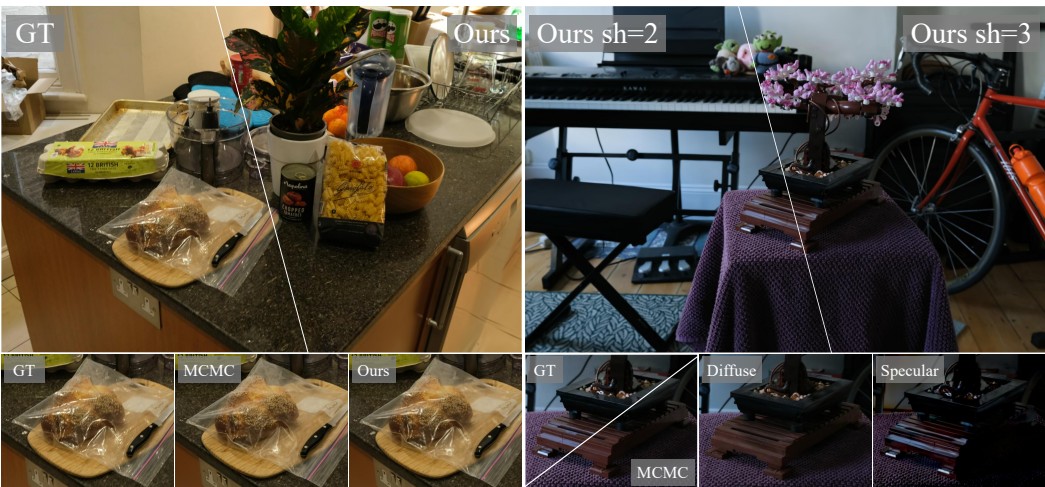

Figure 1: We propose an augmented radiance field, which leverages Gaussian kernels with view-dependent opacity to accurately model specular highlights in the scene (left). It can be seamlessly plugged into existing Gaussian Splatting-based methods as a post enhancement, and notably, even using second-order spherical harmonics (sh=2) is sufficient to capture complex illumination (right).

## Abstract

Due to the real-time rendering performance, 3D Gaussian Splatting (3DGS) has emerged as the leading method for radiance field reconstruction. However, its reliance on spherical harmonics for color encoding inherently limits its ability to separate diffuse and specular components, making it challenging to accurately represent complex reflections. To address this, we propose a novel enhanced Gaussian kernel that explicitly models specular effects through view-dependent opacity. Meanwhile, we introduce an error-driven compensation strategy to improve rendering quality in existing 3DGS scenes. Our method begins with 2D Gaussian initialization and then adaptively inserts and optimizes enhanced Gaussian kernels, ultimately producing an *augmented radiance field*. Experiments demonstrate that our method not only surpasses state-of-the-art NeRF methods in rendering performance but also achieves greater parameter efficiency. Project page at: https://xiaoxinyyx.github.io/augs.

## 1 Introduction

Novel view synthesis, a core task in computer vision and graphics, aims to generate photorealistic images from unobserved viewpoints via 3D scene reconstruction. While Neural Radiance Fields (NeRF) Mildenhall et al. (2020) have significantly advanced rendering quality to this problem, they suffer from slow optimization and rendering. The recent introduction of 3D Gaussian Splatting (3DGS) Kerbl et al. (2023) addresses these issues through an explicit representation based on

---

*Corresponding author.

anisotropic 3D Gaussians and differentiable rasterization, enabling real-time rendering while maintaining high visual fidelity. These capabilities have led to the rapid adoption of 3DGS across a variety of downstream applications, such as autonomous driving Huang et al. (2024b); Zhou et al. (2024), VR Jiang et al. (2024a), and 3D generation Tang et al. (2024); Ren et al. (2023).

However, 3DGS still faces challenges in rendering scenes with complex reflections. Recent advances address this issue from multiple directions. Yu et al. (2024); Yan et al. (2024); Liang et al. (2024); Song et al. (2024) focus on mitigating aliasing artifacts. Xie et al. (2025); Zhang et al. (2025) enhance illumination modeling by integrating environmental lighting. Held et al. (2025); Liu et al. (2025) employ shape-variant primitives beyond smooth Gaussians. Lu et al. (2024); Zhang et al. (2024); Rota Bulò et al. (2024); Kheradmand et al. (2024) propose systematic densification strategies to replace heuristic approaches. Despite these advances, existing Gaussian-based methods still fall short of the synthesis quality achieved by leading NeRF variants, such as Barron et al. (2023), particularly in scenarios requiring the recovery of ultra-high-frequency detail.

3DGS employs third-order spherical harmonics (SH) to model view-dependent radiance, thereby inherently limiting its ability to represent the radiance field. Low-order SH can only capture smooth color transitions and are incapable of reproducing the sharp specular highlights generated by glossy surfaces. Adopting higher-order spherical harmonic coefficients will lead to exponential growth in memory consumption, prolonged training time, and instability in the results of this ill-posed problem. On the other hand, the SH's global definition across the entire sphere limits their efficiency for color encoding, as surface Gaussians are only observable within their outward-facing hemisphere. This mismatch is even more severe in view-dependent cases under directional visibility constraints. Furthermore, SH inherently fail to decouple diffuse and specular components in the rendering equation, complicating material reconstruction and relighting from optimized results.

To compensate for the limitations of spherical harmonics, we propose an *augmented radiance field*, a method for further improving rendering quality based on existing 3DGS scenes, which can be integrated into any splatting-based framework as a *post-enhancement*. Since specular highlights are distributed non-uniformly across the scene (and are usually very sparse), assigning additional parameters to every primitive leads to significant redundancy. Therefore, we introduce an approach that represents the diffuse and specular components using distinct primitives. Specifically, we develop a new Gaussian kernel with view-dependent opacity inspired by Phong shading Phong (1975), which is adaptively integrated into an optimized scene to calibrate regions with reconstruction errors. By adjusting the maximum extent and sharpness of the opacity lobe, this Gaussian kernel is capable of modeling various types of specular highlights, while the complex reflective light distribution on object surfaces can be reconstructed through the superposition of multiple such kernels.

However, allocating and optimizing such enhanced kernels is tricky. We further develop a compensation strategy for scenes that have already been optimized. More specifically, given a pre-trained radiance field of 3DGS, we initialize the enhanced Gaussian kernels using a screen-space loss-based strategy. Initially, 2D Gaussian kernels are sampled in screen space, and only the parameters of these additional 2D Gaussians are optimized to minimize the rendering loss. We then introduce an inverse projection method that maps 2D Gaussians from image space to world space—essentially the inverse of Gaussian Splatting. This densification adaptively allocates Gaussians to regions with high residuals, which typically correspond to areas where the reflective light distribution is complex and challenging to reconstruct using spherical harmonics, and can even capture fine specular spots at the pixel level. Finally, the newly incorporated Gaussians are jointly optimized with the original Gaussians of the scene, resulting in an enhanced radiance field.

Comprehensive experiments are conducted on the widely-used datasets, including Mip-NeRF 360 Barron et al. (2021), Tank & Temples Knapitsch et al. (2017), Deep Blending Hedman et al. (2018), and NeRF Synthetic Mildenhall et al. (2020), using the 3DGS and 3DGS-MCMC Kheradmand et al. (2024) frameworks, respectively. The results indicate that replacing a subset of Gaussians in the scene with Gaussians featuring view-dependent opacity can significantly enhance rendering quality. Remarkably, whether employing second-order or third-order spherical harmonics, our approach consistently outperforms the current state-of-the-art NeRF method Barron et al. (2023), while incurring virtually no loss in real-time rendering speed. Our work demonstrates that Gaussian Splatting-based methods still possesses significant potential for rendering quality improvement.

Our contributions can be summarized as follows:

- We propose an enhanced Gaussian kernel with view-dependent opacity for modeling the specular component. Complex reflections from multiple light sources can be reconstructed by superimposing the new kernels.

- We introduce a novel error-driven method that adaptively and accurately inserts supplementary Gaussians into a scene to improve rendering quality.

- As a Gaussian Splatting-based method, our augmented radiance field outperforms the state-of-the-art NeRF methods while reducing the reliance on higher-order spherical harmonics, achieving both high quality and efficiency.

## 2 RELATED WORK

To align with our goals, we will focus our discussion on novel view synthesis, especially how to further improve the rendering quality of specular surfaces. For applications of NeRF and 3DGS in other domains, we refer readers to Tewari et al. (2022); Fei et al. (2024).

Early attempts on NeRF, such as Zhang et al. (2021); Boss et al. (2021), aim to address this ill-posed problem by decomposing the specular reflectance with the estimated BRDF. Guo et al. (2022) leverages separate neural radiance fields to model the transmitted and reflected components, respectively. Kopanas et al. (2022) proposes a neural warp field that enables efficient point-splatting-based rendering of complex specular effects. Recently, directional encoding has been proposed to improve the rendering quality in the presence of reflections. Their key idea is to model radiance as a direction-dependent function. For example, Verbin et al. (2022) parameterizes the outgoing radiance with the Integrated Directional Encoding (IDE) of the reflective radiance. Ma et al. (2024) further introduces a learnable Gaussian Directional Encoding (GDE) to better model view-dependent effects under near-field lighting conditions. Concurrently, Wu et al. (2024) proposes Neural Directional Encoding (NDE), a novel spatio-spatial parameterization that cone-traces a spatial feature grid to encode near-field reflections and enables modeling multi-bounce reflections. However, since these approaches all build upon NeRF, they also inherit its fundamental limitations, notably the trade-off between high rendering quality and poor computational burden.

Recent advances in Gaussian-based rendering have introduced innovative approaches to handling complex lighting and reflections. Yang et al. (2024) proposes Spec-Gaussian, which utilizes anisotropic spherical Gaussian for modeling view-dependent appearance. Jiang et al. (2024b) enhances neural rendering of 3D Gaussians for reflective scenes by decomposing the appearance into diffuse colors, direct reflections, and a residual term for complex reflections through a novel shading function. Wu et al. (2025) employs deferred shading to decouple scene texture from illumination in Gaussian splatting, achieving more accurate specular modeling. Building on this, Ye et al. (2024) addresses the key challenge of environment map reflections through an improved deferred shading approach, but requires precise surface normals and suffers from discontinuous gradients. Zhu et al. (2024) presents a unified framework that combines SDF-aided Gaussian splatting for relighting optimization with GS-guided SDF enhancement for high-quality geometry reconstruction. Tang & Cham (2024) introduces a continuous local illumination field represented by factorized tensors, enabling optimized incident lighting on 3D Gaussians. Xie et al. (2025) proposes an explicit 3D representation using Gaussian primitives to capture environmental reflections, blending multiple Gaussians during rendering and optimization. Zhang et al. (2025) improves view-dependent effects by directional encoding for 2D Gaussian splatting, leveraging deferred rendering and lighting factorization. Fang et al. (2025) develops a novel framework that integrates NeRF as an assistant to 3DGS, and addresses several limitations of 3DGS while enhancing its performance. Incorporating view-dependent opacity, Vlasic et al. (2003) enables modeling of complex reflective surfaces. Recently, Nowak et al. (2025) enhances the opacity representation of each Gaussian primitive by incorporating a symmetric matrix.

However, existing Gaussian-based approaches still cannot decouple diffuse and specular components in the rendering equation due to the inherent limitations of spherical harmonics. Recently, Liu et al. (2025) advances the representation of material properties by replacing SH with Spherical Beta (SB) functions. Although decoupled modeling is achieved, the beta kernels fall short in representing very complex reflections. Inspired by this work, we propose to enhance 3D Gaussians by explicitly modeling specular components that can be further overlaid to form intricate lobes, thereby providing significantly greater flexibility for handling specular surfaces under complex lighting conditions.

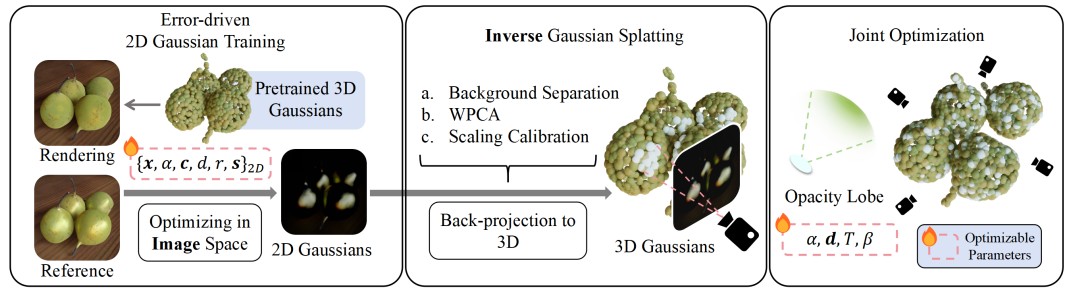

Figure 2: Our **post-enhancement** method for Gaussian Splatting. We begin with image-space re-finement using 2D Gaussians to reconstruct regions exhibiting significant errors (left). Leveraging geometric information from depth maps, we then project these 2D Gaussians into world space (middle). The newly added Gaussians feature optimizable view-dependent opacity and are jointly optimized with existing Gaussians to recover challenging view-dependent color (right).

## 3 AUGMENTED RADIANCE FIELD

We address the limitations of spherical harmonics by equipping Gaussian kernels with a learnable, view-dependent opacity function adapted from the Phong shading paradigm Phong (1975). By compositing multiple such kernels, our method enables high-fidelity reconstruction of complex radiance distributions. Furthermore, we introduce a systematic initialization strategy. As shown in Fig. 2, our method contains three main steps: 2D Gaussians are first sampled and optimized in image space; subsequently, they are geometrically back-projected into 3D space; finally, while keeping most of the original Gaussian parameters fixed, we leverage the differentiable rendering pipeline to jointly refine the supplementary kernels, resulting in an augmented radiance filed. In the following, we first introduce our model of view-dependent opacity, followed by the three steps of our framework.

### 3.1 VIEW-DEPENDENT OPACITY

In the Phong reflection model, the specular component is determined by the viewing direction, the reflection direction, and the material's shininess. Building on this concept, we define a new view-dependent, transparent Gaussian kernel that can model reflectance lobes of various shapes. Specifically, the kernel's opacity is strictly limited to a certain range that peaks when the viewing direction aligns with the lobe's central orientation. Then, the spherical distribution of transparency of this specialized kernel is formulated as:

$$\hat{\alpha}(\theta, \beta, T, \alpha) = \alpha \cdot \left( \frac{\cos\left(\max\left(0, \min\left(\frac{\theta}{T}, \pi\right)\right)\right) + 1}{2} \right)^{\exp(\beta)}. \tag{1}$$

Here, $\alpha$ is the opacity attribute of the 3D Gaussian, $\theta$ denotes the angle between the view direction and the central orientation of the opacity lobe, $T$ controls the angular span of the lobe, and $\beta$ modulates its sharpness. Note that we employ a half-period cosine function to model the opacity lobe, which serves as a variant of the Phong model. Our function exhibits a longer tail and a smoother decay in opacity, resulting in more stable gradients; see Fig. 3 for parameter-dependent variations in lobes. Each supplementary Gaussian kernel adds 5 learnable parameters to the default 3DGS configuration. Crucially, the differentiability of this formulation enables gradient-based optimization of these parameters, as detailed in Section 3.4.

The flexibility of the opacity lobe empowers these Gaussians to approximate specular lobes generated by surfaces with varying roughness under directional illumination. A critical feature is that the transparency of each kernel diminishes to zero as the angular deviation between the viewing direction and the lobe orientation increases, ensuring minimal impact on viewpoints outside the lobe's influence. This property guarantees that enhancing specular components in specific directions incurs no unintended artifacts. By aggregating multiple kernels with heterogeneous opacity distributions, our method effectively reconstructs intricate radiance fields.

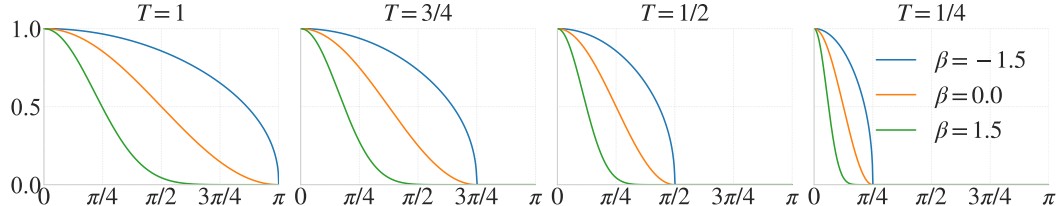

Figure 3: Inspired by classical Phong shading, we model view-dependent opacity with a cosine-weighted function whose shape is controlled by two parameters: $\beta$, which governs the lobe's sharpness, and $T$, which determines its angular extent. Along with the central orientation of the lobe, each new kernel introduces 5 learnable parameters to a standard Gaussian primitive.

### 3.2 ERROR-DRIVEN 2D GAUSSIAN TRAINING

To enable our view-dependent Gaussian kernels to reconstruct complex specular radiance effectively, proper initialization is critical. We propose a novel densification strategy that first samples and optimizes 2D Gaussian kernels in image space, and then backprojects them into world space.

**Optimization in Image Space.** After completing standard 3DGS training, we obtain rendered image sets $\mathcal{I} = \{I_i\}_{i=1}^{M}$ and corresponding depth maps $\mathcal{D} = \{D_i\}_{i=1}^{M}$ for all training views. For each rendered image, we overlay supplementary 2D Gaussians and optimize their parameters using a differentiable rasterizer, which emulates 3DGS's differentiable rendering. The optimization employs the same loss function as 3DGS, with a key difference: we use the pre-rendered image $I_i$ as a fixed background rather than a uniform one. Each 2D Gaussian primitive is defined by parameters:

$$\{\mathbf{x}, \alpha, \mathbf{c}, d, r, \mathbf{s}\}_{2D} \tag{2}$$

where $\mathbf{x}$ denotes image-space coordinates, $\mathbf{s}$ controls the scale along the two principal axes, $r$ specifies the rotation angle between the primary axis and the image y-axis, $\alpha$ represents opacity, $\mathbf{c}$ is the color, and $d$ indicates depth for rendering-order sorting. During optimization, we update each primitive's depth via nearest-neighbor sampling on $D_i$ using its current coordinates.

Prior to training, we predefine the total number of supplementary Gaussians for the scene. Allocation across views is proportional to their respective rendering loss values, with higher-loss views receiving more kernels. During the optimization of 2D kernels for each training image, we periodically augment the kernel set by introducing 20% new primitives at fixed intervals, while pruning those primitives that exhibit excessive transparency or occupy insufficient pixels. When initializing new Gaussian kernels, we adopt a conservative strategy: novel Gaussians are assigned minimal initial opacity, with color sampled from the current rendered image at their mean positions. This design mitigates the perturbation introduced by newly added kernels. The spatial distribution of new kernels is determined through multinomial sampling across pixel coordinates, where the selection probability for each pixel follows:

$$p(u, v) \propto \left[(1 - \lambda_{SSIM}) \mathcal{L}_1(u, v) + \lambda_{SSIM} \mathcal{L}_{SSIM}(u, v)\right]^2. \tag{3}$$

Here $(u, v)$ denotes the image space coordinate. The composite loss combines the L1 and structural similarity metrics, using the same weights as in the original 3DGS paper.

**Depth Map Rendering.** To ensure correct rendering order for 2D Gaussian primitives, depth maps must be rendered for each training image. These maps are subsequently utilized to derive world-space coordinates during projection. Unlike geometric reconstruction methods Dai et al. (2024); Huang et al. (2024a), where 3D Gaussians are approximated as planar surfaces with depths computed via ray-plane intersections, we avoid imposing flattening regularizations. Instead, we adopt a ray-tracing-based approach Moenne-Loccoz et al. (2024) and render depth maps as the positions at which 3D Gaussians attain maximal response along camera rays. As demonstrated in Huang et al. (2024a), median depth, compared to expected depth, yields superior geometric fidelity due to its robustness to outliers. We compute depth values at points where transmittance first drops below 0.5 during rasterization.

3.3 INVERSE GAUSSIAN SPLATTING.

After obtaining the optimized 2D primitives on each training viewpoint, these supplementary Gaussians need to be back-projected into world space to serve as high-quality initializations for new 3D Gaussians. As shown in Fig. 2 (middle), this process involves three key steps: foreground-background separation through point cloud clustering, determination of 3D Gaussian rotations and scales via Weighted Principal Component Analysis (WPCA) Delchambre (2015), and final calibration of the scales.

In object boundary regions, 2D Gaussian kernels often simultaneously cover pixels from both the foreground and background. Without proper grouping and filtering of these regions, subsequent projection would yield suboptimal results. We first compute the 3D coordinates for all pixels covered by a 2D Gaussian kernel, and then perform hierarchical clustering using single-linkage merging Johnson (1967) to group the points. The initial distance threshold for grouping is set proportionally to the depth of the Gaussian center. Priority is given to clusters containing at least three elements that cover the Gaussian center's pixel; if none are available, the largest cluster meeting the minimum three-element requirement is selected. When clustering fails, the distance threshold is gradually increased to retry partitioning. The three-element minimum requirement ensures sufficient data for subsequent computation of the object-space coordinates to derive three mutually orthogonal principal axes and their corresponding directional variances.

To determine the 3D Gaussian parameters, we adopt Weighted PCA, with the 2D Gaussian distribution values as weights for the pixel coordinates. The three principal axes derived from WPCA directly constitute the rotation matrix of the 3D Gaussian kernel, which can be equivalently parameterized as a quaternion, thereby fully defining the rotational attributes. For positional determination, two distinct strategies are employed based on prior grouping outcomes. In the first scenario, where the central pixel successfully passes the filtering process, its world-space coordinates can be directly derived from the depth value and the image-space coordinates of the central pixel. In a typical case where the central pixel fails screening, the depth value is recalibrated as the weighted average of the depths of all filtered pixels, with weights derived from a 2D Gaussian distribution. Subsequently, the central pixel coordinate is combined with the depth value to compute the world-space position.

The scaling factor is the last geometric parameter remaining undetermined for the 3D Gaussian kernels. Although WPCA calculates variances along the three principal axes, directly using these values as the variances for the 3D Gaussians introduces errors. It arises from the finite sampling density on the screen, from points discarded during clustering, and from perspective-projection foreshortening. To minimize back-projection errors, a unified scaling coefficient is applied to all three variances. Revisiting 3DGS's forward projection process, the projected 2D covariance matrix in NDC space for a 3D Gaussian kernel is expressed as:

$$\Sigma_{2D} = JWRS(JWRS)^T \tag{4}$$

where $J \in \mathbb{R}_{2 \times 3}$ denotes the Jacobian matrix as the projective transformation's affine approximation, $W$ represents the world-to-view transformation matrix, and $R$ is the rotation matrix of the Gaussian kernel. The diagonal matrix $S$ encapsulates the Gaussian's scaling factors, defined as $S = k \operatorname{diag}(\sigma_1, \sigma_2, \sigma_3)$, where $\{\sigma_i\}$ are standard deviations computed via WPCA, and $k$ is the target scaling coefficient. Our objective is to minimize the Frobenius norm:

$$\min_k ||\Sigma_{2D} - \hat{\Sigma}_{2D}||_F \tag{5}$$

where $\hat{\Sigma}_{2D}$ corresponds to the target 2D Gaussian's covariance matrix in image space. For convenience, we define $Q := \Sigma_{2D}/k^2$, where all elements of $Q$ are known. Expanding formulation equation 5 yields the solution for $k$ that minimizes the Frobenius norm:

$$k = \sqrt{\frac{\Sigma_{ij} Q_{ij} \hat{\Sigma}_{2D_{ij}}}{||Q||_F^2}}. \tag{6}$$

**Initialization of Opacity Lobe.** Following the projection of 2D Gaussians into world space, the opacity lobe of each primitive requires initialization. The primary orientation is set directly to the unit vector pointing from the Gaussian's centroid to the camera. However, initializing parameters $T$ (controlling the lobe's maximum angular span) and $\beta$ (modulating specular sharpness) presents a

non-trivial challenge. Since the shape of the specular lobe cannot be predetermined, its values are ultimately refined through subsequent multi-view optimization. To ensure that each supplemented kernel is visible to multiple cameras, the initialization of $T$ is based on the spatial distribution of training viewpoints. For each newly added primitive, we first apply view frustum culling to identify the set of cameras $\mathcal{C}_i$ capable of observing it. For each camera in $\mathcal{C}_i$ (excluding the training camera associated with the primitive), the angle between the viewing vector and the lobe orientation is computed. Then, the minimum angle $\bar{\theta}_i$ is selected, and $T$ is initialized as $c\bar{\theta}_i/\pi$, where $c$ is a constant scaling factor. This strategy adaptively initializes $T$ according to the density of surrounding viewpoints. For $\beta$, we simply apply zero initialization.

**Geometry-Independent Parameters.** For parameters unrelated to geometric properties, the base color and opacity of the primitives are initialized to their corresponding 2D primitive counterparts. Higher-order spherical harmonic coefficients undergo zero-initialization.

### 3.4 Joint Optimization

After the projection of all 2D Gaussian kernels into world space, the newly added view-dependent Gaussians undergo joint optimization with pre-existing kernels. During the projection phase, precise parameters for the view-dependent opacity lobes remain temporarily undetermined. To enable multiple projected Gaussians to collectively represent surface-exitant radiance, multi-view constraints are leveraged via photometric error to refine $T$, $\beta$, and opacity orientation across all new primitives. Furthermore, during image-space optimization, the rendered outputs of the original Gaussians are treated as background, assuming they lie behind the newly added Gaussians in depth order. After projection, the supplemented primitives become embedded within the original point cloud, disrupting prior depth hierarchies. Notably, since depth maps are rendered when cumulative transmittance decreases below 0.5, numerous pre-existing Gaussians exhibit shallower depths than the newly added ones. To mitigate over-occlusion of Gaussians, the opacity of the original kernels is made optimizable while freezing their other attributes. For the newly added Gaussians, full parameter optimization is applied to facilitate photometric refinement. We set the number of iterations proportional to the size of the training set to ensure that, across scenarios, each additional Gaussian kernel is sampled approximately the same number of times.

## 4 Experiments

Our method is applicable to most 3DGS-based algorithms. In this work, we mainly evaluated our approach using 3DGS and 3DGS-MCMC Kheradmand et al. (2024) frameworks on a variety of widely used datasets, including Mip-NeRF 360 Barron et al. (2021), Tank & Temples Knapitsch et al. (2017), Deep Blending Hedman et al. (2018), and NeRF Synthetic Mildenhall et al. (2020). We compared our results with state-of-the-art implicit (Zip-NeRF Barron et al. (2023)) and explicit (DBS Liu et al. (2025)) methods. In addition, two recent 3DGS-based methods that specialize in view-dependent reflections, VoD-3DGS Nowak et al. (2025) and Spec-Gaussian Yang et al. (2024), are included for more comprehensive comparisons.

### 4.1 Implementation Detail

Following the 3D Gaussian Splatting framework, we implement a 2D version of the differentiable Gaussian rasterizer. In the image space, we augment each rendered image by setting it as the background and adding 2D Gaussians to further reduce the loss. The ratio of newly added Gaussians to existing ones is fixed at 10%, and they are allocated proportionally to each viewpoint based on the loss values. During optimization of each rendered image, 20% of Gaussians are added incrementally every 200 iterations, and those with insufficient opacity or excessively small scales are pruned. After sufficient kernels were incorporated, an additional 1,000 fine-tuning steps were performed. The projection of 2D Gaussians into world space is executed in parallel on the CPU. During the joint optimization phase, the number of iterations is set to 30 times the size of the training dataset. The Adam optimizer Kingma (2014) is employed to optimize the additional Gaussians, while SparseAdam is exclusively applied to update the opacity of visible original Gaussian primitives. Further details regarding hyperparameter configurations are provided in Appendix A of our supplemental material.

| Method | Mip-NeRF 360 | | | Tanks&Temples | | | Deep Blending | | | NeRF Synthetic | | |
|---|---|---|---|---|---|---|---|---|---|---|---|---|
| | PSNR↑ | SSIM↑ | LPIPS↓ | PSNR↑ | SSIM↑ | LPIPS↓ | PSNR↑ | SSIM↑ | LPIPS↓ | PSNR↑ | SSIM↑ | LPIPS↓ |
| Mip-NeRF 360 | 27.69 | 0.792 | 0.237 | 22.22 | 0.758 | 0.257 | 29.40 | 0.900 | 0.245 | 33.25 | 0.962 | 0.039 |
| Zip-NeRF | 28.54 | 0.828 | 0.189 | - | - | - | - | - | - | 33.10 | 0.971 | 0.031 |
| 3DGS | 27.21 | 0.815 | 0.214 | 23.14 | 0.841 | 0.183 | 29.41 | 0.903 | 0.243 | 33.31 | 0.969 | 0.037 |
| VoD-3DGS[L] | 27.79 | 0.818 | 0.213 | 23.91 | 0.860 | 0.160 | 29.84 | 0.908 | 0.240 | 33.69 | 0.971 | 0.030 |
| Spec-Gaussian | 28.18 | 0.835 | 0.176 | 23.77 | 0.855 | 0.166 | 29.53 | 0.905 | 0.241 | 34.19 | 0.971 | 0.028 |
| 3DGS-MCMC | 28.29 | 0.840 | 0.210 | 24.29 | 0.860 | 0.190 | 29.67 | 0.895 | 0.320 | 33.80 | 0.970 | 0.040 |
| DBS (30k) | 28.60 | 0.844 | 0.182 | 24.79 | 0.868 | 0.148 | 30.10 | 0.910 | 0.240 | 34.64 | 0.973 | 0.028 |
| Ours (3DGS,sh=3) | 28.39 | 0.834 | 0.184 | 23.90 | 0.856 | 0.158 | 29.90 | 0.903 | 0.237 | 34.02 | 0.969 | 0.031 |
| Ours (MCMC,sh=2) | 28.89 | 0.848 | 0.171 | 25.04 | 0.871 | 0.146 | 30.33 | 0.909 | 0.235 | 34.03 | 0.970 | 0.030 |
| Ours (MCMC,sh=3) | 28.96 | 0.849 | 0.170 | 25.06 | 0.872 | 0.144 | 30.22 | 0.909 | 0.236 | 34.35 | 0.971 | 0.029 |

Table 1: Qualitative comparison across Mip-NeRF 360, Tanks&Temples, Deep Blending, and NeRF Synthetic. All scores for the baseline methods are directly taken from their papers, when available. Note that we use the VoD-3DGS variant with higher memory consumption and the anchor-free Spec-Gaussian, as recommended by the authors; for DBS, we adopted the data stopping at 30k iterations for fairness. Our method consistently outperforms state-of-the-art explicit and implicit approaches under the MCMC framework, whether utilizing second- or third-order spherical harmonics.

Fig. 7 further visualizes the rendered Gaussians before and after world-space projection, as well as the L1 distance between the renderings before and after radiance field enhancement.

## 4.2 RESULTS AND ABLATION

**Novel View Synthesis.** Comprehensive evaluations are conducted on 9 Mip-NeRF 360 scenes, 2 Tanks and Temples scenes, 2 Deep Blending scenes, and 8 objects from the NeRF Synthetic dataset, matching 3DGS in resolution and testing protocols. Evaluation metrics include Structural Similarity Index Metric (SSIM), Peak Signal-to-Noise Ratio (PSNR), and Learned Perceptual Image Patch Similarity (LPIPS). While the original 3DGS method lacks control over the total number of Gaussians, our supplementary Gaussians increase the overall count relative to the baseline. Under the 3DGS-MCMC framework, we maintain the total number of Gaussians (original plus newly added) equal to that in the official implementation. For the NeRF Synthetic dataset, we fix the total count of Gaussian primitives at 300,000. We compare rendering quality using second- and third-order spherical harmonics (SH).

Results summarized in Table 1 demonstrate that our Gaussian supplementation strategy significantly enhances rendering quality across existing GS scenes. It is worth noting that, within the MCMC framework, although second-order SH (*i.e.,* sh=2) reduce the parameter count by 21 per primitive compared to third-order SH (sh=3), they still achieve comparable rendering quality (see Table 15 in Appendix D for memory comparisons). Both approaches outperform the state-of-the-art implicit method Barron et al. (2023), demonstrating that our method facilitates the adaptation of Gaussian splatting to low-end hardware platforms. Compared with DBS Liu et al. (2025), the state-of-the-art explicit approach, our method outperforms it on real-world datasets but slightly lags behind on the NeRF Synthetic dataset. This discrepancy stems from the simplicity of synthetic environments, in which each scene contains only a single object and exhibits limited illumination and material variation. In contrast, our method leverages these advantages in complex real-world scenarios through a flexible strategy for superposing opacity lobes. Note that, to ensure a fair comparison, we used the data at 30k iterations from DBS Liu et al. (2025), as their full model monitored test-set metrics to achieve the best performance, whereas most other baselines used a fixed number of iterations. Compared with specular-aware methods, VoD-3DGS and Spec-Gaussian, our method achieves better results on almost all datasets. We further compare the training time, memory usage, and FPS in the supplementary material.

Fig. 4 and Fig. 8 (Appendix D) present qualitative analysis, which reveals restored specular highlights in challenging regions and marked improvements for indoor scenes with complex material compositions. Additionally, we observe that supplementary Gaussians effectively enhance under-reconstructed distant regions in outdoor scenarios. See Appendix D for qualitative metrics per scene.

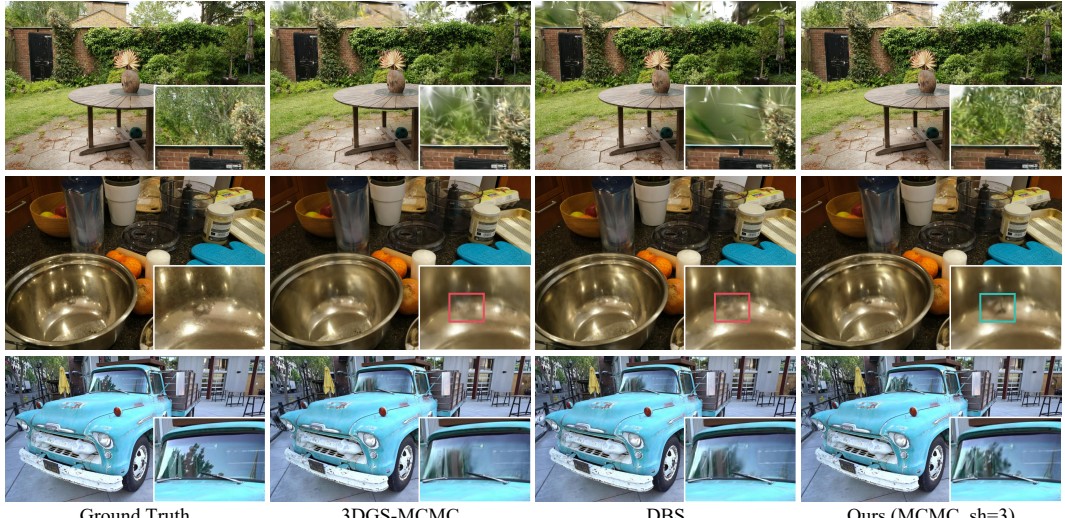

| Ground Truth | 3DGS-MCMC | DBS | Ours (MCMC, sh=3) |

Figure 4: Qualitative comparison across real captured scenes. Benefiting from the newly designed Gaussian kernel with view-dependent transparency and the rendering loss-driven initialization, our approach surpasses all state-of-the-art explicit and implicit methods on real-world datasets.

Table 2: Impact of the ratio of enhanced Gaussians with the total primitive No. unchanged.

| Ratio | Stage | Mip-NeRF 360 | | Tanks&Temples | |
|---|---|---|---|---|---|
| | | PSNR↑ | SSIM↑ | PSNR↑ | SSIM↑ |
| 5% | basic | 28.31 | 0.846 | 24.48 | 0.867 |
| | augmented | 28.88 | 0.849 | 24.95 | 0.872 |
| 10% | basic | 28.33 | 0.845 | 24.53 | 0.867 |
| | augmented | 28.96 | 0.849 | 25.06 | 0.872 |
| 15% | basic | 28.30 | 0.845 | 24.52 | 0.867 |
| | augmented | 28.94 | 0.849 | 24.99 | 0.871 |

Table 3: Ablation study on Mip-NeRF 360 dataset. Optimizing the orientation and shape of the opacity lobe achieves render quality unattainable by solely supplementing with Gaussians without view-dependent opacity.

| Methods | PSNR↑ | SSIM↑ | LPIPS↓ |
|---|---|---|---|
| 3DGS-MCMC | 28.33 | 0.845 | 0.176 |
| Ours(MCMC), w/o opacity lobe | 28.45 | 0.847 | 0.171 |
| Ours(MCMC), $T = 0.5$ | 28.60 | 0.848 | 0.170 |
| Ours(MCMC), $\beta = 0$ | 28.93 | 0.849 | 0.170 |
| Ours(MCMC), full model | 28.96 | 0.849 | 0.170 |

**Analysis on Primitive Ratio.** To evaluate the impact of the ratio of additional Gaussian kernels on the reconstruction, we conducted experiments using the MCMC framework. Specifically, we maintained a constant total number of Gaussians while varying the ratio between newly added and original kernels. Tests on Mip-NeRF 360 and Tanks & Temples (Table 2) have demonstrated optimal rendering quality when supplementary Gaussians constitute 10% of the total count of primitives.

**Ablation Study.** Table 3 summarizes the rendering quality evaluation under various configurations. We exclusively selected the Mip-NeRF 360 dataset for ablation studies due to its consistent photometric properties across viewpoints and minimal presence of overexposed pixels. This choice allows the algorithm to focus on reconstructing surface exitant radiance in complex scenes rather than compensating for exposure variations. Disabling the opacity lobe of supplementary Gaussians yielded only a 0.12dB PSNR improvement. Fixing the opacity lobe coverage to $\pi/2$ while retaining view-dependent opacity increased PSNR by 0.27dB (from 28.33 dB baseline). Optimizing the $T$ parameter further exhibits significant improvements. Best performance is achieved with full optimization of opacity lobes' parameters.

To validate the superiority of our enhanced Gaussian kernels over spherical harmonics, we conducted experiments using high-order spherical harmonics within the 3DGS framework. Results in Table 4 demonstrate that increasing color representation parameters yields no substantial performance gains, indicating significant parameter redundancy in 3DGS, which is also evidenced in 3DGS compression work Niedermayr et al. (2024); Papantonakis et al. (2024).

Table 4: Increasing the order of spherical harmonics in 3DGS does not bring substantial improvement in rendering quality.

| SH order | Mip-NeRF 360 | | | |
| | PSNR↑ | SSIM↑ | LPIPS↓ | Mem (MB)↓ |
|---|---|---|---|---|
| 3 | 27.47 | 0.814 | 0.217 | 608 |
| 4 | 27.55 | 0.813 | 0.217 | 887 |

Table 5: Our method outperforms Spherical Beta color modeling under high-frequency illumination and specular surface conditions.

| Methods | Glossy surface | | Mirror-like surface | |
| | PSNR↑ | SSIM↑ | PSNR↑ | SSIM↑ |
|---|---|---|---|---|
| DBS (sb=2) | 41.70 | **0.993** | 29.48 | **0.941** |
| Ours (MCMC, sh=3) | **42.33** | **0.993** | **29.73** | **0.941** |

As shown previously, our method demonstrates marginally inferior performance compared to the Spherical Beta function-based color modeling approach Liu et al. (2025) on the NeRF Synthetic dataset, primarily due to the dataset's prevalence of rough surfaces and limited high-frequency environmental illumination, where two Spherical Beta functions per primitive suffice for accurate reconstruction. However, when high-frequency components dominate the radiance field, the adaptive superposition characteristics of our opacity lobe exhibit superior efficacy. To validate that, we constructed a synthetic shiny dataset under extreme illumination conditions. As reported in Table 5, we demonstrated improved reconstruction fidelity proportional to the complexity of surface exitant radiance distributions. We attribute our superiority to the flexibility of the enhanced Gaussians in stacking opacity lobes of different kernels. More analyses are provided in Appendix B.

### 4.3 DISENTANGLING DIFFUSE AND SPECULAR COMPONENTS

Following DBS Liu et al. (2025), our augmented radiance fields with enhanced Gaussian primitives can also achieve explicit diffuse and specular component separation. It should be noted that all scenes in this work are rendered in the sRGB color space – a perceptually uniform space where pixel values do not linearly correlate with radiance intensity. Prior to component disentanglement, we transform rendered images to the linear radiance space.

Our separation protocol proceeds as follows: First, we render the linear-space image $I_{\text{sh}_0}$ using only the diffuse color of original Gaussians. Subsequently, the enhanced linear-space image $I_{\text{aug}}$ is rendered with our augmented radiance field. The diffuse component is then computed as $I_d = \min(I_{\text{sh}_0}, I_{\text{aug}})$, while the specular component comes from $I_s = I_{\text{aug}} - I_d$. As shown in Fig. 1, Fig. 9, and Fig. 10 in Appendix D (results converted back to sRGB for visualization), our method successfully disentangles white reflections on bonsai, specular highlights on Lego blocks, solar reflections on truck scene pavements, and metallic sheens on drum kits. Notably, limitations arise from the low dynamic range of training images, where overexposed pixels introduce separation artifacts due to irreversible radiance saturation.

### 5 CONCLUSION

In this work, we present an enhanced Gaussian Splatting method that equips Gaussian kernels with view-dependent opacity. By leveraging our error-driven compensation strategy that carefully inserts enhanced primitives, we achieved state-of-the-art 3D Gaussian Splatting across various datasets. Moreover, as a post-processing step for 3DGS, our approach is plug-and-play compatible with most existing 3DGS frameworks and may improve their rendering quality. Another promising direction is to further subdivide specialized Gaussian kernels to represent finer-grained rendering components.

### REPRODUCIBILITY STATEMENT

The source code and dataset for this paper will be made publicly available. Implementation details can be found in Section 4.1, while Appendix A presents more comprehensive hyperparameter settings. Appendix C provides pseudocode for inverse Gaussian splatting.

ACKNOWLEDGEMENTS

This work was supported in parts by the National Key R&D Program of China (2024YFB3908500, 2024YFB3908502), NSFC (U21B2023), Guangdong Basic and Applied Basic Research Foundation (2023B1515120026), and Scientific Development Funds from Shenzhen University.

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

## A  FULL IMPLEMENTATION DETAILS

We optimize the 2D Gaussian primitives in image space using the Adam Kingma (2014) optimizer, where the learning rate for image-space coordinates $\mathbf{x}_{2D}$ is set to 0.001 multiplied by the mean of the image dimensions (width and height), with learning rates of 0.01 for color $\mathbf{c}_{2D}$, 0.02 for opacity $\alpha_{2D}$, 0.001 for scale $\mathbf{s}_{2D}$, and 0.02 for rotation angle $r_{2D}$. Every 200 iterations, 20% of Gaussian primitives are added, while those with opacity below 0.005 or covering fewer than 25 pixels are pruned. For supplementary Gaussians, the initial value of $T$ is set to $7\bar{\theta}_i/pi$, constrained within the range of 0.01 to 1.0.

Prior to projecting 2D Gaussian kernels into world space, the sampling pixels are clustered to remove foreground or background pixels. Upon obtaining world space coordinates corresponding to pixels covered by Gaussians, these coordinates are classified using the single linkage merging algorithm Sibson (1973), which operates by iteratively merging the closest cluster pairs based on the minimum distance between any two points from distinct clusters. The process continues until either a single cluster remains or a termination threshold is satisfied. The initial termination threshold is determined by the depth value of the Gaussian kernel's central pixel, specifically by computing the world-space distance $\delta d$ between two adjacent pixels at this depth, with the threshold set to $5 \cdot \delta d$. In cases of clustering failure, the threshold is multiplied by 1.5 and the procedure is repeated. Pseudocode is provided in Appendix C.

During the final joint optimization stage, learning rates are configured as follows: 0.000016 for position, 0.001 for diffuse color, 0.001/20 for higher-order spherical harmonic coefficients, 0.02 for opacity, 0.002 for scaling, 0.0005 for rotation, 0.001 for opacity lobe orientation $\mathbf{d}$, 0.0002 for opacity lobe width $T$, and 0.002 for sharpness $\beta$. Concurrently, SparseAdam is employed to optimize the opacity of original Gaussian primitives visible in the current viewpoint. All experiments were conducted on a workstation equipped with an Intel i9-13900K CPU and an NVIDIA RTX 4090 GPU. On the Mip-NeRF 360 dataset, the training and projection of 2D Gaussians averages approximately 15 minutes per scene, as detailed in Table 6.

Table 6: The time consumption for the post-enhancement stage on the Mip-NeRF 360 and NeRF Synthetic datasets is reported as follows. It should be noted that using multiple GPUs can significantly accelerate this process, as the 2D-stage training operates independently on each image.

| Dataset | Mip-NeRF 360 | NeRF Synthetic |
|---|---|---|
| 2D Training and Projection | 742 s | 178 s |
| Joint Optimization | 168 s | 32 s |

## B  ANALYSIS ON OPACITY LOBE

While DBS Liu et al. (2025) replaces spherical harmonics with spherical beta functions in their formulation, our approach fundamentally differs by applying generalized Phong shading to opacity modulation rather than radiance modeling. The implementation of view-dependent opacity introduces greater stability and flexibility in radiance reconstruction, because the opacity lobe can decouple each individual Gaussian kernel's contribution to the final exitant radiance. This distinction is empirically validated through a controlled experiment simulating three overlapping Gaussian kernels in Fig. 5, where we compare two configurations: 1) optimization with spherical beta functions governing color distributions while maintaining fixed opacity values, versus 2) our methodology employing view-dependent opacity modulation with fixed color magnitudes. Under identical opacity settings ($\alpha = 0.5$ for all kernels). The spherical beta approach requires exponentially increasing color magnitudes for subsurface Gaussians to compensate for occlusion effects from overlying kernels (Fig. 5, top-left), creating instability because kernel depth ordering varies dynamically during optimization and view changes. Conversely, our method (second row) achieves target radiance distributions through adaptive opacity variations while maintaining constant color magnitudes, demonstrating remarkable resilience to permutation of depth sorting. This comparative analysis conclusively demonstrates that our Gaussian kernel formulation exhibits substantially greater stability and flexibility than DBS's design.

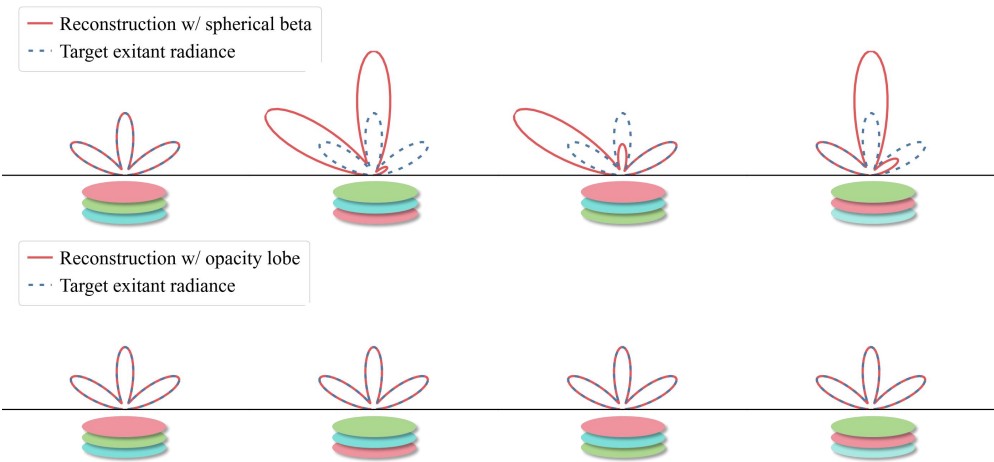

Figure 5: Comparison between Spherical Beta functions in DBS Liu et al. (2025) and opacity lobe. The three Gaussians in the first row model color using three differently oriented spherical beta functions, with maximum function values of 1, 2, and 4, all having an opacity of 0.5. The kernels in the second row use three differently oriented opacity lobes, with the same maximum amplitude and identical color intensity. Our method is more stable and flexible in reconstructing outgoing radiance, as it is less affected by the ordering of Gaussian kernels and has greater numerical stability.

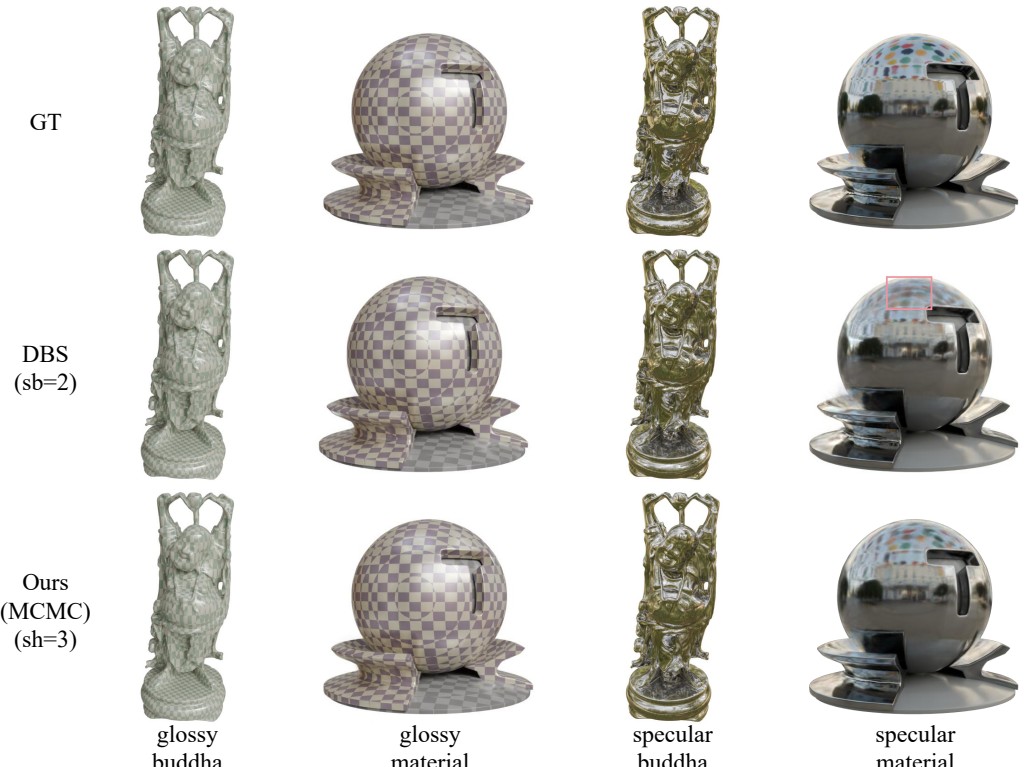

Figure 6: Qualitative comparison on our synthetic shiny dataset rendered using environment maps with complex lighting conditions.

To validate that, we built a synthetic shiny dataset under extreme illumination conditions, as shown in Fig. 6. We demonstrated improved reconstruction fidelity that scales with the complexity of surface exitant radiance distributions, which supports the superiority of our enhanced Gaussians in stacking opacity lobes across different kernels.

```python
 1  def inverse_splatting(gaussian_2d, depth_map, camera):
 2      pixels_world = pixels_covered(gaussian_2d, depth_map, camera)
 3      # Step 1. Foreground-background separation
 4      threshold = sample_tex(depth_map, gaussian_2d.center)
 5      do:
 6          threshold = threshold * A
 7          groups = cluster_points(threshold, pixels_world)
 8      # The PCA algorithm requires at least three input data points.
 9      while groups.min_size() < 3
10      # Step2. WPCA
11      filtered_points, INCLUDE_CENTER = prime_group(groups)
12      rotation, scaling = WPCA(filtered_points)
13      # Gaussian center included in the filtered points
14      if INCLUDE_CENTER:
15          depth = sample_tex(depth_map, gaussain_2d.center)
16      else:
17          depth = weighted_mean(filtered_points.depth, filtered_points.
                weight)
18      position = depth_to_world(depth, gaussian_2d.center, camera)
19      # Step3. Scaling calibration
20      Q = project(rotation, scaling, position, camera)
21      k = sqrt((Q * gaussian_2d.cov).sum() / frobenius_norm(Q) ** 2)
22      scaling = k * scaling
23      return rotation, scaling, position
```

Listing 1: Pseudocode for our inverse splatting algorithm.

## C  INVERSE GAUSSIAN SPLATTING

By leveraging the rendered depth map and camera parameters, we project the 2D Gaussians from image space into world space. Fig. 7 compares the rendered Gaussians from a single training image before and after projecting them into world space. This process is implemented through three steps to ensure reasonable and accurate projection results, with pseudocode provided in List 1.

## D  ADDITIONAL COMPARISONS AND RESULTS

Fig. 8 presents more qualitative analysis of our full model. Table 8, 9, and 10 present the rendering metrics of each scene in Table 1 of the main paper, enhanced with our supplementary kernels. For outdoor scenes in Mip-NeRF 360, our method achieves approximately 0.1 dB PSNR improvement, while indoor scenes with complex lighting and material variations exhibit 1-2 dB gains. Tables 11, 12 and 13 quantify performance variations before and after radiance field augmentation, while Table 14 and 15 presents a comparison of the memory footprint for the scenes in the Mip-NeRF 360 dataset. The proposed opacity lobe introduces minimal overhead, requiring only 5 parameters per instance, with $\sim 10\%$ of primitives storing these additional parameters. This design ensures negligible memory impact compared to baseline implementations.

Fig. 9 shows more results of our diffuse-specular decomposition. To investigate the performance, we further conducted a qualitative comparison with DBS's Liu et al. (2025) decomposition in Fig. 10. Because the dataset lacks ground-truth specular data, quantitative metrics cannot be reported. Visually, our method produces a specular component comparable to that of DBS.

We compare the training time, memory usage, and FPS with Spec-Gaussian Yang et al. (2024) in Table 7. Since Spec-Gaussian employs a neural network for real-time rendering, it significantly impacts the FPS. This also affects the scene reconstruction speed. Our method requires less memory and achieves a much higher FPS.

Table 7: Analysis of training time, memory usage, and FPS. "Total time" represents the combined duration of the original scene reconstruction and post-processing enhancement.

| Method | Mip-NeRF 360 | | | NeRF Synthetic | | |
|---|---|---|---|---|---|---|
| | Total(Post-proc.) time | Memory | FPS | Total(Post-proc.) time | Memory | FPS |
| Spec-Gaussian | 44.8 min | 882 MB | 43 | 12.0 min | 75 MB | 136 |
| Ours (3DGS,sh=3) | 27.9(15.0) min | 691 MB | 110 | 6.0(3.4) min | 68 MB | 446 |
| Ours (MCMC,sh=2) | 31.9(14.9)min | 467 MB | 96 | 7.0(3.2)min | 44 MB | 434 |
| Ours (MCMC,sh=3) | 25.3(15.2)min | 722 MB | 90 | 7.9(3.5) min | 68 MB | 414 |

Table 8: Quantitative results of rendering quality per scene on Mip-NeRF 360 dataset.

| | metrics | bicycle | flowers | garden | stump | treehill | room | counter | kitchen | bonsai | mean |
|---|---|---|---|---|---|---|---|---|---|---|---|
| Zip-NeRF | PSNR↑ | 25.80 | 22.40 | 28.20 | 27.55 | 23.89 | 32.65 | 29.38 | 32.50 | 34.46 | 28.54 |
| | SSIM↑ | 0.769 | 0.642 | 0.860 | 0.800 | 0.681 | 0.925 | 0.902 | 0.928 | 0.949 | 0.828 |
| | LPIPS↓ | 0.208 | 0.273 | 0.118 | 0.193 | 0.242 | 0.196 | 0.185 | 0.116 | 0.173 | 0.189 |
| 3DGS-MCMC | PSNR↑ | 26.15 | 22.12 | 28.16 | 27.80 | 23.31 | 32.48 | 29.51 | 32.27 | 32.88 | 28.29 |
| | SSIM↑ | 0.81 | 0.64 | 0.89 | 0.82 | 0.66 | 0.94 | 0.92 | 0.94 | 0.95 | 0.84 |
| | LPIPS↓ | 0.18 | 0.31 | 0.10 | 0.19 | 0.29 | 0.25 | 0.22 | 0.14 | 0.22 | 0.21 |
| DBS (30k) | PSNR↑ | 26.04 | 22.44 | 28.15 | 27.56 | 23.49 | 32.83 | 30.36 | 32.61 | 33.90 | 28.60 |
| | SSIM↑ | 0.804 | 0.650 | 0.882 | 0.815 | 0.676 | 0.941 | 0.930 | 0.939 | 0.957 | 0.844 |
| | LPIPS↓ | 0.169 | 0.308 | 0.096 | 0.184 | 0.293 | 0.168 | 0.154 | 0.110 | 0.156 | 0.182 |
| Ours 3DGS sh=3 | PSNR↑ | 25.85 | 21.81 | 28.21 | 27.30 | 23.00 | 32.39 | 30.48 | 32.24 | 34.26 | 28.39 |
| | SSIM↑ | 0.793 | 0.633 | 0.881 | 0.796 | 0.660 | 0.931 | 0.924 | 0.936 | 0.954 | 0.834 |
| | LPIPS↓ | 0.174 | 0.294 | 0.093 | 0.186 | 0.289 | 0.183 | 0.166 | 0.110 | 0.165 | 0.184 |
| Ours MCMC sh=2 | PSNR↑ | 26.31 | 22.34 | 28.35 | 27.97 | 23.49 | 33.32 | 30.64 | 33.01 | 34.60 | 28.89 |
| | SSIM↑ | 0.815 | 0.658 | 0.887 | 0.827 | 0.680 | 0.942 | 0.928 | 0.940 | 0.957 | 0.848 |
| | LPIPS↓ | 0.158 | 0.273 | 0.088 | 0.159 | 0.262 | 0.169 | 0.160 | 0.108 | 0.161 | 0.171 |
| Ours MCMC sh=3 | PSNR↑ | 26.32 | 22.44 | 28.36 | 27.90 | 23.46 | 33.45 | 30.82 | 33.15 | 34.78 | 28.96 |
| | SSIM↑ | 0.814 | 0.663 | 0.887 | 0.825 | 0.680 | 0.942 | 0.930 | 0.941 | 0.958 | 0.849 |
| | LPIPS↓ | 0.158 | 0.270 | 0.086 | 0.161 | 0.259 | 0.170 | 0.158 | 0.106 | 0.159 | 0.170 |

## E LIMITATIONS

The back-projection of 2D Gaussians into world space relies on depth maps of training cameras, where high-quality depth estimation could improve projection accuracy. However, this work does not incorporate geometric constraints or deep neural networks to generate refined depth maps, resulting in projected Gaussian kernels that do not accurately conform to object surfaces. Additionally, the designed two-stage optimization strategy prolongs scene reconstruction time, which could be alleviated by integrating the initialization of newly added Gaussians with the training process of the original scene. Furthermore, although a fixed ration between added and original Gaussians is maintained to accommodate most scenarios, this approach may introduce redundancy in scenes dominated by diffuse materials. Adaptively determining the number of additional Gaussians per scene remains a promising direction for future improvements.

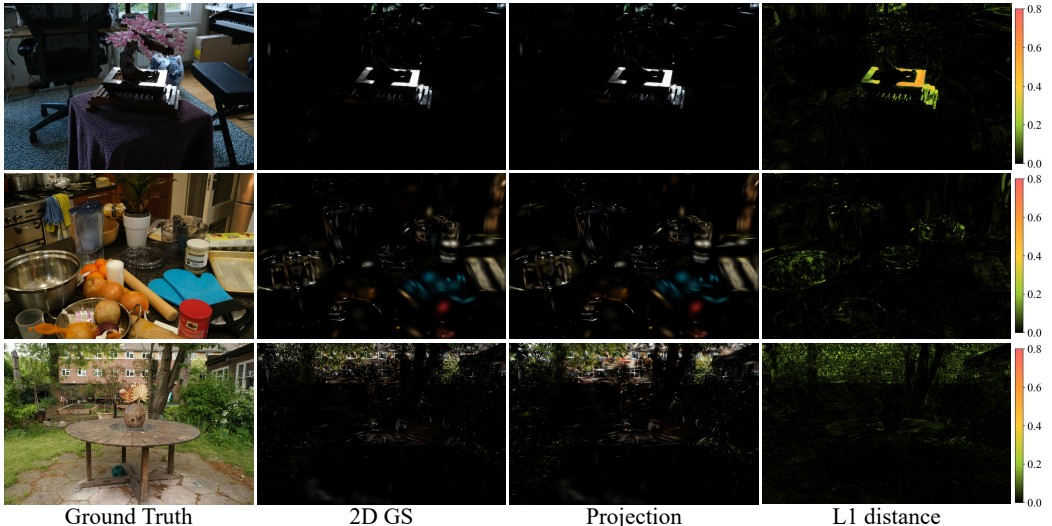

Figure 7: Projection results of 2D Gaussians. We present the optimization results of the 2D primitives in image space along with the rendered outputs after back-projection into world space. The 4th column images display the L1 distance between the renderings before and after enhancement. The additional Guassians can also help restore the background reconstruction for outdoor scenes.

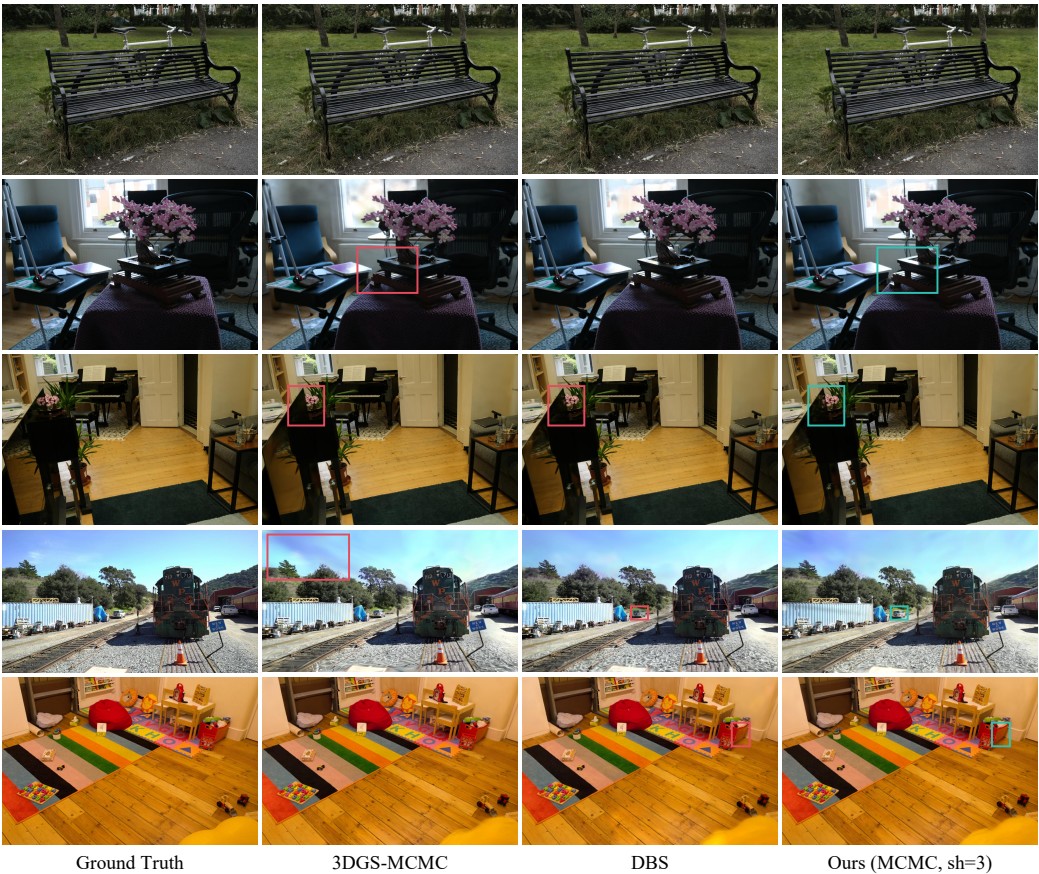

Figure 8: Qualitative analysis across real-captured scenes.

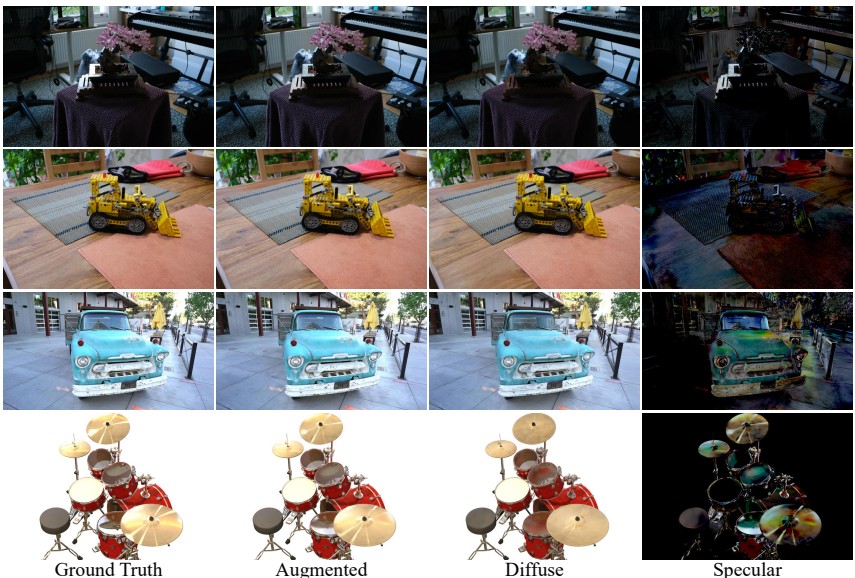

Figure 9: Diffuse specular decomposition. We convert both the diffuse color rendering of the original Gaussians and the enhanced rendering into linear color space to decouple the diffuse and specular components.

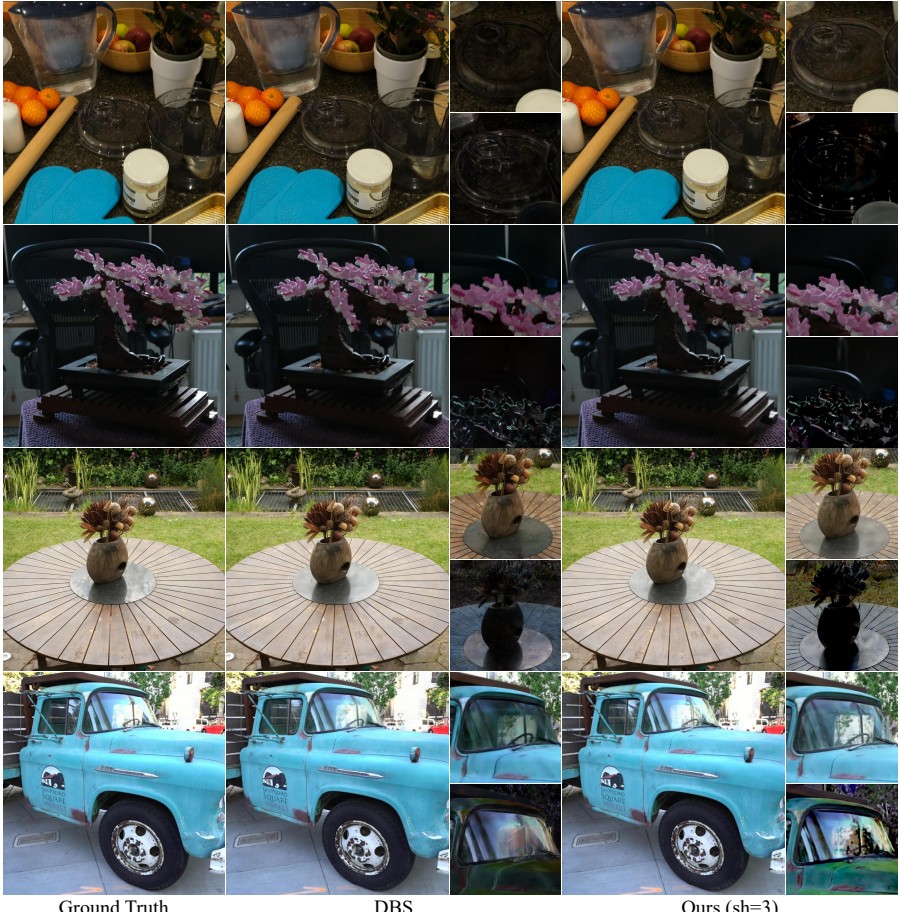

Figure 10: Qualitative comparison with DBS on diffuse-specular decomposition. The top-right and bottom-right sub-figures display the diffuse and specular components, respectively.

Table 9: Quantitative results of rendering quality per scene on Tanks and Temples and Deep Blending dataset.

| | metrics | Tanks and Temples | | | Deep Blending | | |
|---|---|---|---|---|---|---|---|
| | | truck | train | mean | drjohnson | playroom | mean |
| Zip-NeRF | PSNR↑ | - | - | - | - | - | - |
| | SSIM↑ | - | - | - | - | - | - |
| | LPIPS↓ | - | - | - | - | - | - |
| 3DGS-MCMC | PSNR↑ | 26.11 | 22.47 | 24.29 | 29.00 | 30.33 | 29.67 |
| | SSIM↑ | 0.89 | 0.83 | 0.86 | 0.89 | 0.90 | 0.89 |
| | LPIPS↓ | 0.14 | 0.24 | 0.19 | 0.33 | 0.31 | 0.32 |
| DBS (30k) | PSNR↑ | 26.44 | 23.13 | 24.79 | 29.39 | 30.82 | 30.10 |
| | SSIM↑ | 0.897 | 0.839 | 0.868 | 0.908 | 0.912 | 0.910 |
| | LPIPS↓ | 0.110 | 0.185 | 0.147 | 0.240 | 0.240 | 0.240 |
| Ours 3DGS sh=3 | PSNR↑ | 25.71 | 22.09 | 23.90 | 29.31 | 30.50 | 29.90 |
| | SSIM↑ | 0.886 | 0.826 | 0.856 | 0.902 | 0.905 | 0.903 |
| | LPIPS↓ | 0.131 | 0.185 | 0.158 | 0.236 | 0.237 | 0.237 |
| Ours MCMC sh=2 | PSNR↑ | 26.78 | 23.30 | 25.04 | 29.90 | 30.77 | 30.33 |
| | SSIM↑ | 0.899 | 0.842 | 0.871 | 0.904 | 0.914 | 0.909 |
| | LPIPS↓ | 0.110 | 0.182 | 0.146 | 0.234 | 0.236 | 0.235 |
| Ours MCMC sh=3 | PSNR↑ | 26.86 | 23.27 | 25.06 | 29.85 | 30.58 | 30.22 |
| | SSIM↑ | 0.900 | 0.844 | 0.872 | 0.904 | 0.913 | 0.909 |
| | LPIPS↓ | 0.106 | 0.181 | 0.144 | 0.235 | 0.237 | 0.236 |

Table 10: Quantitative results of rendering quality per scene on NeRF synthetic dataset.

| | metrics | chair | drums | ficus | hotdog | lego | materials | mic | ship | mean |
|---|---|---|---|---|---|---|---|---|---|---|
| Zip-NeRF | PSNR↑ | 34.84 | 25.84 | 33.90 | 37.14 | 34.84 | 31.66 | 35.15 | 31.38 | 33.10 |
| | SSIM↑ | 0.983 | 0.944 | 0.985 | 0.984 | 0.980 | 0.969 | 0.991 | 0.929 | 0.971 |
| | LPIPS↓ | 0.017 | 0.050 | 0.015 | 0.020 | 0.019 | 0.032 | 0.007 | 0.091 | 0.031 |
| 3DGS-MCMC | PSNR↑ | 36.51 | 26.29 | 35.07 | 37.82 | 36.01 | 30.59 | 37.29 | 30.82 | 33.80 |
| | SSIM↑ | 0.99 | 0.95 | 0.99 | 0.99 | 0.98 | 0.96 | 0.99 | 0.91 | 0.97 |
| | LPIPS↓ | 0.02 | 0.04 | 0.01 | 0.02 | 0.02 | 0.04 | 0.01 | 0.12 | 0.04 |
| DBS (30k) | PSNR↑ | 36.74 | 26.78 | 36.75 | 38.85 | 37.12 | 31.12 | 37.66 | 32.13 | 34.64 |
| | SSIM↑ | 0.990 | 0.958 | 0.990 | 0.988 | 0.985 | 0.966 | 0.994 | 0.910 | 0.973 |
| | LPIPS↓ | 0.010 | 0.033 | 0.010 | 0.015 | 0.014 | 0.032 | 0.005 | 0.103 | 0.028 |
| Ours 3DGS sh=3 | PSNR↑ | 35.73 | 26.59 | 35.71 | 38.21 | 36.39 | 30.74 | 37.13 | 31.63 | 34.02 |
| | SSIM↑ | 0.988 | 0.955 | 0.987 | 0.985 | 0.982 | 0.961 | 0.993 | 0.896 | 0.969 |
| | LPIPS↓ | 0.011 | 0.036 | 0.012 | 0.020 | 0.016 | 0.036 | 0.006 | 0.113 | 0.031 |
| Ours MCMC sh=2 | PSNR↑ | 36.05 | 26.39 | 35.47 | 38.34 | 36.44 | 30.69 | 37.27 | 31.59 | 34.03 |
| | SSIM↑ | 0.989 | 0.955 | 0.987 | 0.987 | 0.985 | 0.963 | 0.993 | 0.904 | 0.970 |
| | LPIPS↓ | 0.011 | 0.038 | 0.012 | 0.016 | 0.016 | 0.035 | 0.006 | 0.106 | 0.030 |
| Ours MCMC sh=3 | PSNR↑ | 36.35 | 26.57 | 35.87 | 38.72 | 36.67 | 31.11 | 37.79 | 31.68 | 34.35 |
| | SSIM↑ | 0.989 | 0.956 | 0.988 | 0.988 | 0.985 | 0.965 | 0.994 | 0.902 | 0.971 |
| | LPIPS↓ | 0.011 | 0.036 | 0.011 | 0.016 | 0.015 | 0.033 | 0.006 | 0.106 | 0.029 |

Table 11: Quantitative analysis of rendering quality improvements after enhancement per scene on Mip-NeRF 360 dataset.

| Method | Metrics | bicycle | flowers | garden | stump | treehill | room | counter | kitchen | bonsai | Mean |
|---|---|---|---|---|---|---|---|---|---|---|---|
| Ours (3DGS, sh=3) | ΔPSNR↑ | 0.110 | -0.067 | 0.307 | 0.089 | 0.168 | 0.720 | 1.230 | 0.806 | 1.765 | 0.570 |
| | ΔSSIM↑ $(10^{-3})$ | 3.26 | 5.17 | 2.84 | 4.77 | 3.19 | 2.77 | 5.49 | 2.11 | 5.25 | 3.87 |
| | ΔLPIPS↓ $(10^{-3})$ | -5.09 | -28.6 | -3.00 | -9.09 | -14.9 | -5.04 | -8.08 | -3.16 | -7.23 | -9.35 |
| Ours (MCMC, sh=2) | ΔPSNR↑ | 0.227 | 0.007 | 0.397 | 0.212 | 0.085 | 0.967 | 1.622 | 0.976 | 2.225 | 0.746 |
| | ΔSSIM↑ $(10^{-3})$ | 4.44 | 3.39 | 4.83 | 3.25 | 2.96 | 4.52 | 8.66 | 3.14 | 6.39 | 4.62 |
| | ΔLPIPS↓ $(10^{-3})$ | -5.58 | -12.5 | -5.24 | -4.62 | -7.04 | -6.97 | -11.7 | -4.94 | -7.93 | -7.38 |
| Ours (MCMC, sh=3) | ΔPSNR↑ | 0.174 | 0.003 | 0.239 | 0.093 | 0.079 | 0.978 | 1.376 | 0.798 | 1.975 | 0.635 |
| | ΔSSIM↑ $(10^{-3})$ | 3.24 | 2.75 | 3.48 | 2.24 | 1.60 | 3.88 | 6.60 | 2.59 | 5.11 | 3.50 |
| | ΔLPIPS↓ $(10^{-3})$ | -6.26 | -12.3 | -4.57 | -4.11 | -6.91 | -6.35 | -9.66 | -4.33 | -6.83 | -6.81 |

Table 12: Quantitative analysis of rendering quality improvements after enhancement per scene on NeRF synthetic dataset.

| Method | Metrics | chair | drums | ficus | hotdog | lego | materials | mic | ship | Mean |
|---|---|---|---|---|---|---|---|---|---|---|
| | $\Delta$PSNR$\uparrow$ | 0.130 | 0.239 | 0.132 | 0.080 | 0.160 | 0.137 | 0.146 | 0.113 | 0.142 |
| Ours (3DGS, sh=3) | $\Delta$SSIM$\uparrow$ ($10^{-4}$) | 1.82 | 4.45 | 1.97 | 2.37 | 3.57 | 4.38 | 1.50 | 6.20 | 3.28 |
| | $\Delta$LPIPS$\downarrow$ ($10^{-4}$) | -0.724 | -5.99 | -1.73 | -5.69 | -3.56 | -3.32 | -1.25 | -6.17 | -3.55 |
| | $\Delta$PSNR$\uparrow$ | 0.265 | 0.288 | 0.419 | 0.231 | 0.364 | 0.400 | 0.464 | 0.235 | 0.333 |
| Ours (MCMC, sh=2) | $\Delta$SSIM$\uparrow$ ($10^{-4}$) | 3.42 | 4.74 | 6.54 | 5.85 | 6.45 | 12.5 | 4.36 | 3.86 | 5.97 |
| | $\Delta$LPIPS$\downarrow$ ($10^{-4}$) | -4.92 | -13.0 | -5.97 | -7.44 | -6.47 | -11.5 | -3.69 | -13.9 | -8.36 |
| | $\Delta$PSNR$\uparrow$ | 0.174 | 0.251 | 0.252 | 0.150 | 0.271 | 0.185 | 0.278 | 0.114 | 0.209 |
| Ours (MCMC, sh=3) | $\Delta$SSIM$\uparrow$ ($10^{-4}$) | 2.38 | 4.72 | 3.70 | 2.57 | 4.88 | 5.31 | 2.51 | 3.04 | 3.64 |
| | $\Delta$LPIPS$\downarrow$ ($10^{-4}$) | -4.15 | -9.39 | -4.50 | -5.79 | -5.41 | -6.68 | -2.32 | -9.15 | -5.92 |

Table 13: Quantitative analysis of rendering quality improvements after enhancement per scene.

| Method | Metrics | Tanks and Temples | | | Deep Blending | | |
|---|---|---|---|---|---|---|---|
| | | truck | train | Mean | drjohnson | playroom | Mean |
| | $\Delta$PSNR$\uparrow$ | 0.348 | 0.356 | 0.352 | 0.169 | 0.059 | 0.114 |
| Ours (3DGS, sh=3) | $\Delta$SSIM$\uparrow$ ($10^{-3}$) | 2.90 | 3.04 | 2.97 | 1.29 | 0.537 | 0.915 |
| | $\Delta$LPIPS$\downarrow$ ($10^{-3}$) | -7.44 | -3.34 | -5.39 | -5.96 | -6.31 | -6.14 |
| | $\Delta$PSNR$\uparrow$ | 0.460 | 0.522 | 0.491 | 0.256 | 0.126 | 0.191 |
| Ours (MCMC, sh=2) | $\Delta$SSIM$\uparrow$ ($10^{-3}$) | 4.07 | 6.18 | 5.12 | 2.17 | 1.28 | 1.72 |
| | $\Delta$LPIPS$\downarrow$ ($10^{-3}$) | -7.40 | -7.87 | -7.63 | -6.14 | -7.87 | -7.01 |
| | $\Delta$PSNR$\uparrow$ | 0.442 | 0.618 | 0.530 | 0.231 | 0.238 | 0.235 |
| Ours (MCMC, sh=3) | $\Delta$SSIM$\uparrow$ ($10^{-3}$) | 3.62 | 6.58 | 5.10 | 1.83 | 1.35 | 1.59 |
| | $\Delta$LPIPS$\downarrow$ ($10^{-3}$) | -7.45 | -7.00 | -7.22 | -5.80 | -8.08 | -6.94 |

Table 14: Quantitative results of memory footprint per scene on Mip-NeRF 360 dataset.

| Method | Metrics | bicycle | flowers | garden | stump | treehill | room | counter | kitchen | bonsai | Mean |
|---|---|---|---|---|---|---|---|---|---|---|---|
| 3DGS | Primitives ($10^6$) | 4.8 | 3.0 | 4.1 | 4.5 | 3.3 | 1.4 | 1.2 | 1.5 | 1.2 | 2.8 |
| | Memory (MB) | 1081 | 675 | 928 | 1017 | 732 | 316 | 263 | 328 | 275 | 624 |
| DBS | Primitives ($10^6$) | 6.0 | 3.0 | 5.0 | 4.5 | 3.5 | 1.5 | 1.5 | 1.5 | 1.5 | 3.1 |
| | Memory (MB) | 618 | 309 | 515 | 464 | 361 | 155 | 155 | 155 | 155 | 320 |
| Ours (3DGS, sh=3) | Primitives ($10^6$) | 5.3 | 3.3 | 4.5 | 5.0 | 3.6 | 1.5 | 1.3 | 1.6 | 1.3 | 3.0 |
| | Memory (MB) | 1198 | 748 | 1029 | 1128 | 811 | 350 | 291 | 363 | 305 | 691 |

Table 15: Quantitative results of memory footprint per scene on Mip-NeRF 360 dataset.

| | | bicycle | flowers | garden | stump | treehill | room | counter | kitchen | bonsai | Mean |
|---|---|---|---|---|---|---|---|---|---|---|---|
| Method | Primitives ($10^6$) | 5.9 | 3.3 | 5.2 | 4.8 | 3.7 | 1.5 | 1.2 | 1.8 | 1.3 | 3.2 |
| 3DGS-MCMC | | 1328 | 743 | 1170 | 1069 | 833 | 338 | 270 | 405 | 293 | 716 |
| Ours (MCMC, sh=2) | Memory (MB) | 865 | 484 | 763 | 697 | 543 | 220 | 176 | 264 | 191 | 467 |
| Ours (MCMC, sh=3) | | 1338 | 748 | 1179 | 1077 | 839 | 340 | 272 | 408 | 295 | 722 |

