# OpenReview forum: "Augmented Radiance Field: A General Framework for Enhanced Gaussian Splatting"
_ICLR.cc/2026/Conference — ICLR 2026 Poster_

### Official Review · Reviewer_eJHV · 2025-10-28

**Soundness:** 3
**Presentation:** 3
**Contribution:** 3
**Rating:** 4
**Confidence:** 4

**Summary:**

The paper presents Augmented Radiance Field (ARF), a plug-and-play framework that enhances 3D Gaussian Splatting (3DGS) for realistic view synthesis. While 3DGS achieves real-time rendering, its spherical harmonic color model struggles with specular reflections and material separation. ARF introduces a Gaussian kernel with view-dependent opacity to explicitly model specular effects. It further proposes an error-driven refinement that inserts and optimizes additional Gaussians in high-error regions. Experiments on multiple benchmarks show that ARF outperforms state-of-the-art NeRF and 3DGS variants.

**Strengths:**

1. The paper introduces a new Gaussian kernel with view-dependent opacity that effectively models specular reflections and high-frequency lighting effects.

2. The proposed 2D-to-3D error compensation mechanism adaptively adds and optimizes supplementary Gaussians in challenging regions.

3. Extensive experiments across several benchmarks demonstrate consistent improvements over state-of-the-art NeRF and 3DGS methods, even with lower-order spherical harmonics.

**Weaknesses:**

1. Since the method refines pre-trained 3DGS scenes rather than learning end-to-end, it may accumulate suboptimal biases from the base model and depend on prior scene quality. The image-space optimization and 2D-to-3D projection introduce extra computation and memory costs compared to standard 3DGS.

2. The paper does not analyze training stability or convergence behavior when optimizing new Gaussians, especially in highly reflective or complex lighting conditions.

3. The rendering quality heavily depends on the tuning of the opacity lobe parameters beta; the paper does not provide an analysis of how these parameters influence optimization.

**Questions:**

See weakness.

---

> ### Author Response · Authors · 2025-11-17
> **To reviewer eJHV**
>
> We sincerely thank you for acknowledging our strengths. Here are our responses to your questions.
>
> ## 1. May accumulate suboptimal biases
> Our method does not accumulate errors. On the contrary, as a post-processing procedure, it is designed to **correct errors** generated during the reconstruction of the base scene. We considered developing an end-to-end version of this approach, but doing so would **compromise its generalizability**. In its current form, our pipeline can be applied to any 3DGS-based method to further improve reconstruction quality. For users pursuing photorealistic scene reconstruction, we provide a **plug-and-play** module. Experimental data demonstrate that our method already achieves state-of-the-art performance on real-world datasets. Integrating the Gaussian supplementation step directly into the 30,000-iteration training would force other frameworks to re-tune hyperparameters, reducing usability.
>
> ## 2. Training stability
> Thank you for the valuable suggestion. After introducing new Gaussians via back-projection, the rendering quality may experience a temporary decline. However, the reconstruction quality rapidly surpasses that of the original scene after 10N optimization steps, where N denotes the size of the reference image set. In subsequent training steps, the loss value decreases very slowly and gradually plateaus. We found that 30N steps serves as an appropriate stopping point, being sufficiently robust to handle all the scenarios.
>
>
> ## 3. Heavily depends on the tuning of the opacity lobe parameters beta
> In Table 3 of the paper, ablation studies show that the rendering quality primarily depends on the width `T` of the opacity lobe, rather than `beta`. In fact, both parameters can adjust the lobe width, but `T` has a more significant effect. When projecting 2D Gaussians from different views into world space, their lobes often overlap substantially. Optimizing `T` efficiently mitigates this issue.

---

> ### Author Response · Authors · 2025-11-28
> **Request for Status Update Following Rebuttal**
>
> Dear Reviewer eJHV,
>
> We understand that you are exceptionally busy, and we are very grateful for the time and effort you invested in our work.
>
> We were very encouraged to see that Reviewer zmRh, after reading our point-by-point responses, raised the score and gave a very positive assessment. We believe our rebuttal has adequately addressed the issues raised and hope it has also clarified your concerns.
>
> Given the positive development with one reviewer's evaluation, could you please provide a brief update on the current status?
>
> Thank you again for your consideration and guidance throughout this process. We look forward to hearing from you.
>
> Best,
>
> The authors

---

### Official Review · Reviewer_LiBR · 2025-10-29

**Soundness:** 2
**Presentation:** 2
**Contribution:** 2
**Rating:** 4
**Confidence:** 4

**Summary:**

This paper addresses limitations of 3D Gaussian Splatting (3DGS) in handling complex reflections due to its reliance on spherical harmonics for color encoding. The authors propose an enhanced Gaussian kernel that models specular effects via view-dependent opacity, combined with an error-driven compensation strategy to improve rendering quality. Their method starts with 2D Gaussian initialization, followed by adaptive insertion and optimization of enhanced kernels to produce an augmented radiance field.

**Strengths:**

This paper proposes a novel post-enhancement method for Gaussian splatting based on the Phong shading model, aiming to improve the modeling of view-dependent color.

**Weaknesses:**

- The paper leverages geometric information from depth maps of a pre-trained 3DGS and back-projects the screen-space 2D Gaussians into world space. However, due to the limitation of low-order spherical harmonics in 3DGS, the reconstructed scene geometry tends to be quite poor. As is well known, more accurate geometry modeling typically leads to more reliable view-dependent color estimation. Unfortunately, the paper does not provide any comparison between the optimized depth results and those from the pre-trained 3DGS. For instance, in the garden scene of the Mip-NeRF 360 dataset, the flat tabletop region could have been used as a clear example for such a comparison.
- The proposed method does not fundamentally address the limitation of using spherical harmonics in 3DGS for separating diffuse and specular components. For example, Spec-Gaussian leverages anisotropic spherical Gaussians to better model view-dependent appearance. Instead of optimizing a pre-trained 3DGS, it might be more meaningful to design a more effective approach for modeling view-dependent effects directly, since the use of low-order SH inherently reflects a trade-off between rendering speed and quality.

The paper has some typos:
1. Line 101, "prossesses"
2. Line 155-156, "view-dependently"
3. Line 188, "...  reconstructs intricate radiance field"

**Questions:**

See Weaknesses

---

> ### Author Response · Authors · 2025-11-17
> **To reviewer LiBR**
>
> We sincerely appreciate your valuable feedback and hope our response effectively solves your concerns.
>
> ## 1. Geometric accuracy and comparison of depth
> Within the 3DGS framework, improved geometric accuracy does not necessarily lead to improved rendering quality. Recently, several works have focused on optimizing and enhancing geometry for Gaussian splatting, including the introduction of depth priors (from pre-trained large models), multi-view consistency, and planarity constraints. However, these efforts have not yielded significant improvements in rendering quality. On the contrary, improving the rendering quality of 3DGS remains a very challenging problem.
>
> Anyway, we are happy to make a comparison between the optimized depth results and those from the pre-trained 3DGS.
>
> ## 2. Separating diffuse and specular components
> Fully decoupling diffuse and specular components is a challenging task. In Spec-Gaussian, the low-order spherical harmonic rendering result is combined with a view-dependent effect modeled by an MLP. Even with this explicit modeling of view-dependent effects from the beginning, they did **not** achieve a complete decoupling of diffuse and specular components, and the final performance is still **lower** than our method. Similarly, DBS also directly models view-dependent effects but has not surpassed our metrics. The advantage of our enhancement approach is that we **push the rendering quality of 3DGS to its limits** as much as possible. For applications pursuing photorealistic rendering, our method provides a significant boost. Moreover, our enhancement module is **plug-and-play** and can be integrated into any training framework to deliver noticeable improvements. We also considered integrating this enhancement process into the 30,000-iteration training phase rather than applying it post-reconstruction. However, doing so would compromise the generalizability of the method, as others would need to readjust hyperparameters when adopting it in their own frameworks, reducing convenience. Finally, as suggested by reviewer zmRh, our method also possesses the potential **application to alternative primitives** beyond Gaussians. We are optimistic about its future.

---

> ### Author Response · Authors · 2025-11-28
> **Request for Status Update Following Rebuttal**
>
> Dear Reviewer LiBR,
>
> We understand that you are exceptionally busy, and we are very grateful for the time and effort you invested in our work.
>
> We were very encouraged to see that Reviewer zmRh, after reading our point-by-point responses, raised the score and gave a very positive assessment. We believe our rebuttal has adequately addressed the issues raised and hope it has also clarified your concerns.
>
> Given the positive development with one reviewer's evaluation, could you please provide a brief update on the current status?
>
> Thank you again for your consideration and guidance throughout this process. We look forward to hearing from you.
>
> Best,
>
> The authors

---

### Official Review · Reviewer_mqPT · 2025-10-31

**Soundness:** 3
**Presentation:** 3
**Contribution:** 2
**Rating:** 2
**Confidence:** 4

**Summary:**

The paper augments 3D Gaussian Splatting (3DGS) with a view-dependent opacity lobe to better reproduce specular effects. It introduces a post-enhancement pipeline that (i) detects high-residual regions in image space and optimizes sparse 2D Gaussians there, (ii) back-projects these into 3D via an “inverse splatting” procedure, and (iii) jointly optimizes the enhanced set with the original scene. The module is positioned as plug-and-play for existing 3DGS pipelines and aims to improve quality with modest runtime overhead.

**Strengths:**

1. The two-stage 2D residual fixing, inverse splatting, and joint optimization could improve the final rendering quality.
2. The proposed method could be served as a post-hoc module on top of the 3DGS-based methods.

**Weaknesses:**

1. The primary concern is the design choice to separate diffuse and specular components across different primitives. In real scenes, most surfaces exhibit a mixture of both (with mirrors as a special extreme), and prior work that models these components within unified primitives enables shared optimization, coherent regularization, and cleaner support for inverse rendering and relighting. By contrast, the proposed error-driven initialization of dedicated “specular” primitives breaks this unification, ties specular capacity to residual patterns rather than material properties, and risks uneven coverage and missed interactions. As a result, the approach sacrifices some of the interpretability and downstream utility.
2. Lobe-based models for specular appearance (notably spherical Gaussians) are well established in rendering and have been adopted within 3DGS frameworks (e.g., SpecGaussian [A]). Beyond these, there are various prior works tailored to specular scenes, including methods with broader applications such as inverse rendering and relighting. The manuscript would benefit from direct comparisons with these baselines to demonstrate the performance of the proposed design.
3. Some statements about "SH inherently fails to decouple diffuse and specular components" on line 070 conflict with the definitions of the specular and diffuse components on line 138. SH0 could capture view-dependent colors, with higher orders capturing view-dependent colors. Some explanations about why SH inherently fails to decouple diffuse and specular components are needed.

[A] Spec-Gaussian: Anisotropic View-Dependent Appearance for 3D Gaussian Splatting

**Questions:**

1. How to compute the gradients when performing the optimization in image space? As described in 209-210, the depth is updated via the nearest-neighbor sampling during optimization. However, since the emulated 3DGS differentiable rendering (described on line 201) requires depth sorting (in both forward and backward pass), this depth updating strategy will influence the gradients computation. Specifically, how many rasterization passes are performed during an optimization step? when to perform the depth updating, like before or after updating other parameters?
2. What are the end-to-end training time and rendering FPS, compared with baselines (including baselines designed for specular scenes, such as SpecGaussian)?

---

> ### Author Response · Authors · 2025-11-17
> **To reviewer mqPT**
>
> We sincerely thank you for your valuable feedback and hope that our response addresses your concerns effectively.
>
> ## 1. Using different primitives and more comparisons
> For your first two concerns, regarding the use of different primitives and additional comparisons (such as Spec-Gaussian), please refer to our response to common issues. Our method surpasses Spec-Gaussian on all benchmarks, and we believe that using different primitives is a promising new direction for improving the rendering quality of 3DGS.
>
> ## 2. Why SH fails to decouple diffuse and specular components
> Regarding why SH cannot decouple diffuse and specular components, a simple explanation is that both diffuse and specular components must remain **non-negative** over the spherical domain, and SH can be viewed as a spherical Fourier decomposition. Except for the zero-order coefficient, which satisfies non-negativity, higher-order SH components contain negative values over the sphere (the integral of any higher-order basis function over the sphere is zero). Therefore, one cannot simply assign the zero-order spherical harmonics to represent the diffuse component and higher-order components to represent specular reflection.
>
> However, we still aim to leverage spherical harmonics combined with our newly introduced primitive to estimate diffuse and specular components, even though these estimates may not be physically exact. To ensure the non-negativity of both diffuse and specular components, when rendering the diffuse component, we do not directly use the zero-order spherical harmonic output. Instead, we take the **minimum value** between the zero-order rendering and the enhanced rendering result (see Line 471).
>
> ## 3. How to compute the gradients in image space
> **After each update** of the 2D Gaussian parameters (screen coordinates, opacity, color, scale, and rotation), we sample the precomputed depth map at the current screen coordinates of the 2D Gaussian's center point and directly update its depth value with the sampled result. The depth parameter carried by the 2D Gaussian serves **only one purpose** during the 2D rasterization stage: determining the **rendering order** of the Gaussians. The depth parameter does not affect the size of the 2D Gaussian in image space, which can be understood as using an **orthographic projection** rather than a perspective projection at this stage. Consistent with the original 3DGS implementation, each optimization step performs only one forward pass and one backward pass.
>
> ## 4. Training time and rendering FPS
> Please see our response to common issues.

---

> > ### Comment · Reviewer_mqPT · 2025-11-28
> > **Thank you for your response and clarification.**
> >
> > Thank you for your response and clarification.
> >
> > Regarding the use of different primitives, I recognize the potential benefits of the proposed method in terms of maximizing rendering quality. Under this objective, I think both the anchor-based and anchor-free Spec-Gaussian variants appear to be natural and appropriate baselines.
> >
> > 1. To more clearly demonstrate the advantages of explicit geometry, could you provide training-time and FPS comparisons between Spec-Gaussian and your method when both are built on top of the original 3DGS backbone, thereby isolating the contribution of the MCMC component?
> > 2. In addition, have you considered re-enabling the densification module on a pre-trained 3DGS model? This could offer further insight into the relative benefits of your explicit-geometry design versus improved optimization of standard 3DGS.
> >
> > I would be inclined to raise my overall score if these experiments can more clearly substantiate the advantages of the proposed method.

---

> ### Author Response · Authors · 2025-12-04
> **Additional experiments provided by the authors**
>
> Thank you for your response. Regarding the first question, we have updated the table, which can be viewed in the common issues (part 1) section. When using 3DGS as the backbone, the performance in terms of training time, memory usage, and FPS **is better than that of MCMC**.
>
> As for the second question, we attempted to further optimize the pre-trained 3DGS model for 10,000 steps. The first 5,000 steps involved enabling the heuristic densification strategy of 3DGS, while the subsequent 5,000 steps continued to optimize other parameters. The results indicate that this approach may add more primitives to the scene but does not improve the rendering quality. This outcome aligns with our intuition—additional densification operations cannot overcome the inherent limitations of the algorithm. There will always be primitives occupying excessively large spaces that fail to meet the splitting conditions, resulting in Gaussian blur.
>
>
> | Method     | Mip-   | NeRF  | 360   |        | \|  | NeRF  | Synthetic |       |        |
> | ---------- | ------ | ----- | ----- | ------ | --- | ----- | --------- | ----- | ------ |
> |            | Mem    | PSNR↑ | SSIM↑ | LPIPS↓ |     | Mem   | PSNR↑     | SSIM↑ | LPIPS↓ |
> | 3DGS (30k) | 605 MB | 27.82 | 0.830 | 0.194  |     | 61 MB | 33.87     | 0.968 | 0.032  |
> | 3DGS (40k) | 667 MB | 27.75 | 0.829 | 0.192  |     | 61 MB | 33.88     | 0.968 | 0.031  |

---

### Official Review · Reviewer_zmRh · 2025-10-31

**Soundness:** 4
**Presentation:** 4
**Contribution:** 4
**Rating:** 6
**Confidence:** 5

**Summary:**

This paper proposes a strategy to enhance the under-reconstructed regions of 3DGS. The method begins by learning a 3DGS model to fit the given multi-view images. However, due to limitations in the original densification and optimization processes of 3DGS, certain regions may not be reconstructed effectively.

To address this, the method identifies under-reconstructed regions by comparing the rendered images with the GT images. An error-weighted adaptive strategy is then employed to sample 2DGS on each view screen, targeting these problematic regions. Using the original rendered images as backgrounds, the 2DGS in each view is optimized to correct the image and align it with the GT.

Although 2DGS is defined in screen space, the order of the splats is determined by the depth of the nearest rendered pixel. After optimization, clustering is performed for each 2D Gaussian to identify inlier pixels. These inliers are back-projected into 3D space, and a weighted PCA is applied to analyze their Gaussian distribution. The resulting 3D points are then directly used to initialize the supplemented 3DGS.

Finally, these supplemented 3DGS points, along with the opacity of the original 3DGS, are jointly optimized to fit the input images, resulting in a more accurate 3D reconstruction.

**Strengths:**

1. The paper addresses the under-reconstruction problem of 3D Gaussian Splatting (3DGS) directly and effectively by employing a smart solution: correcting problematic regions through 2D fitting and back-projection. The entire process carefully accounts for various factors in the pipeline that could lead to artifacts, making the approach both robust and practical. By "reintegrating missing details into 3D space," the method achieves improved reconstruction. The idea is plausible, insightful, and highly meaningful.

2. The rendered images produced by this method significantly outperform the baselines. Highlighted and specular regions are reconstructed with remarkable accuracy. This improvement is largely attributed to the proposed directional decaying opacity, which enables the method to better capture and fit high-frequency details in these challenging areas.

**Weaknesses:**

The primary weakness of the paper lies in the limited comparison with baselines. The main baseline used in the study is deformable beta splatting, which explores alternative primitive representation functions to replace Gaussians. However, two additional relevant baselines are worth considering for comparison:

1. **Spec-Gaussian**: Spec-Gaussian introduces a more advanced lighting function to replace SHs, enabling better handling of commonly observed anisotropic appearances. It achieves superior results in specular and highlight regions, making it closely related to the view-dependent opacity proposed in the paper.

2. **AbsGS**: AbsGS leverages the absolute gradient norm accumulation as a metric to densify 3DGS. By using the absolute gradients from the 2D screen, it can effectively identify under-reconstructed regions, thereby improving the densification of Gaussians in problematic areas. This is closely related to the enhancement-on-the-bad idea proposed in the paper.

**Questions:**

- I hold a positive attitude toward the proposed method; however, my main concern is the limited comparison with the closely related baselines mentioned, which weakens the overall persuasiveness of the paper. I would be willing to raise the score to accept if these comparisons are included. :)

- The proposed method does not appear to be restricted to a specific representation, such as 3DGS. Beyond 3DGS, other flexible primitives, such as Deformable Beta Splatting (DBS), 3D Convex Splatting, and Deformable Radial Kernel (DRK), can also efficiently capture sharp regions and varying boundaries. These primitives could be incorporated into the 2D fixing step as well. Among them, DRK stands out as a strict projection-based representation, making it much easier to back-project and achieve accurate depth rendering. I suggest including a discussion on the potential application of the method to these more flexible primitives, as it could provide valuable insights and inspiration for future research.

---

> ### Author Response · Authors · 2025-11-17
> **To reviewer zmRh**
>
> We sincerely thank you for your positive feedback and support. It's truly encouraging and heart-warming.
>
> ## 1. More comparisons
> The suggested comparisons with Spec-Gaussian and AbsGaussian are summarized in our response to common issues. Our method outperforms these approaches across all benchmarks.
>
> ## 2. Potential application to more flexible primitives
> Thank you for such a constructive suggestion. Yes, the approach we proposed—optimizing in image space followed by back-projection—can be applied to methods that do not use 3D Gaussians as primitives, in which case we **only need to redesign the back-projection algorithm** (i.e., the second step of our method). We are quite optimistic about this direction. Firstly, as we have validated in the paper, our strategy enhances the reconstruction of specular highlights. Secondly, it can serve as an auxiliary densification strategy. While existing methods typically perform densification by splitting or cloning primitives to increase density in sparse regions, our method differs in that the process of introducing new primitives is **content-aware**. Guided by the ground truth image, we provide **high-quality initialization** for new primitives, which helps reduce inefficient primitives and accelerates convergence. We hope to further explore the potential of content-aware densification in future work.

---

> ### Comment · Reviewer_zmRh · 2025-11-25
>
> Thank you to the authors for their response. I have also reviewed the comments from other reviewers and would like to address some points of disagreement regarding the weaknesses raised:
>
> 1. **Reprojection using trained 3DGS damages the end-to-end process, making it ineffective**: I disagree with this criticism. As proposed in the original 3DGS framework, the initial point cloud is initialized from SfM points, which are based on feature matching and are not entirely reliable. The training process of 3DGS offers a more effective initialization compared to SfM, thereby helping to mitigate issues in problematic regions.
>
> 2. **Separating diffuse and specular effects into different primitives is suboptimal**: When training a radiance field, the primary goal is to model radiance accurately rather than focusing on material separation, as 3DGS was not originally designed for relighting purposes. The ultimate objective is to achieve high-quality novel view synthesis. Therefore, the design choice, whether diffuse and specular components are represented separately or even predicted by unexplainable MLPs, should not be a point of concern.
>
> With the provided comparisons against Spec-Gaussian and AbsGS, the results clearly demonstrate better performance. Based on this, I have decided to raise my score. I hold a positive attitude toward the paper and am confident in its contributions. Best of luck to the authors!

---

> ### Author Response · Authors · 2025-11-25
> **Gratitude from the Author**
>
> We would like to express our sincere gratitude for your strong support and insightful comments. From our experience, improving the rendering quality of 3DGS is really tough. Your understanding of our work's contribution and your strong defense against other reviewers' criticisms are crucial and so encouraging. We sincerely thank you for your consistent support and fair judgment.
>
> Thank you once again.

---

### Author Response · Authors · 2025-11-17
**Response to Common issues (part 1)**

We would like to thank the reviewers for the valuable feedback on our work. We are pleased to see that the reviewers affirmed the effectiveness of our method, especially encouraged by the praise that described it as "**a smart solution**" and noted that "**The idea is plausible, insightful, and highly meaningful.**" Below, we focus on the common issues in this response. Other personal questions are replied to in the individual comments.

## 1. More comparisons
The first and most important question is the **comparison with other baselines**, particularly **Spec-Gaussian** [1] (Reviewer zmRh also mentioned AbsGS [2]). To ensure a fair comparison (in terms of training time and FPS), for the key baseline Spec-Gaussian, we downloaded their code and reproduced the experiments on our machine. Our reproduced metrics are slightly lower than those reported in their paper. We list both results in the table (distinguished as "official" and "repro."). For AbsGS, we directly adopted the data from their paper. The comparisons are summarized below (in bold for the best and underlined for the second best).

| Method | Mip- | NeRF | 360 | \|  | | T&T | | \|  | Deep | Blending | | \|  | NeRF | Synthetic | |
| --- | :--: | :-: | :-: | :-: | :-: | :-: | :-: | :-: | :-: | :-: | :-: | :-: | :-: | :-: | :-: |
| | PSNR↑ | SSIM↑ | LPIPS↓ | | PSNR↑ | SSIM↑ | LPIPS↓ | | PSNR↑ | SSIM↑ | LPIPS↓ | | PSNR↑ | SSIM↑ | LPIPS↓ |
| Spec-Gaussian (official) | 28.18 | 0.835 | 0.176 | | - | - | - | | - | - | - | | $\underline{34.19}$ | $\textbf{0.971}$ | $\textbf{0.028}$ |
| Spec-Gaussian (repro.)   | 27.97 | 0.834 | 0.176 | | 23.77 | 0.855 | 0.166 | | 29.53 | $\underline{0.905}$ |        0.241        |     |        34.02        | $\underline{0.970}$ |  $\textbf{0.028}$   |
| AbsGS                    |        27.49        |        0.820        |        0.191        |     |        23.73        |        0.853        |        0.162        |     |        29.67        |        0.902        | $\underline{0.236}$ |     |          -          |          -          |          -          |
| Ours (3DGS,sh=3)         |        28.39        |        0.834        |        0.184        |     |        23.90        |        0.856        |        0.158        |     |        29.90        |        0.903        |        0.237        |     |        34.02        |        0.969        |        0.031        |
| Ours (MCMC,sh=2)         | $\underline{28.89}$ | $\underline{0.848}$ | $\underline{0.171}$ |     | $\underline{25.04}$ | $\underline{0.871}$ | $\underline{0.146}$ |     |  $\textbf{30.33}$   |  $\textbf{0.909}$   |  $\textbf{0.235}$   |     |        34.03        | $\underline{0.970}$ |        0.030        |
| Ours (MCMC,sh=3)         |  $\textbf{28.96}$   |  $\textbf{0.849}$   |  $\textbf{0.170}$   |     |  $\textbf{25.06}$   |  $\textbf{0.872}$   |  $\textbf{0.144}$   |     | $\underline{30.22}$ |  $\textbf{0.909}$   | $\underline{0.236}$ |     |  $\textbf{34.35}$   |  $\textbf{0.971}$   | $\underline{0.029}$ |

As shown above, Spec-Gaussian's reconstruction quality lags behind ours to some extent. Furthermore, their code utilizes different parameter configurations for real and non-real scenes, as well as indoor and outdoor scenes, whereas our method employs a unified hyperparameter setting.

Below is the comparison of training time, memory usage, and FPS. We use "total time"​ to represent the combined duration of the original scene reconstruction and the post-processing enhancement.

| Method               |      Mip-NeRF 360       |        |     | \|  |     NeRF Synthetic      |       |     |
| ---------------------- | :---------------------: | :----: | :-: | :-: | :---------------------: | :---: | :-: |
|                        | Total (Post-proc.) time |  Mem   | FPS |     | Total (Post-proc.) time |  Mem  | FPS |
| Spec-Gaussian (repro.) |        44.8 min         | 882 MB | 43  |     |        12.0 min         | 75 MB | 136 |
| Ours (3DGS,sh=3) | 32.5 (19.5) min | 671 MB | 116 | | 11.3 (8.0) min | 68 MB | 292 |
| Ours (MCMC,sh=2)      |     36.5 (19.4) min     | 467 MB | 90  |     |     12.4 (7.6) min      | 44 MB | 279 |
| Ours (MCMC,sh=3)      |     40.6 (20.6) min     | 722 MB | 86  |     |     12.8 (7.6) min      | 68 MB | 270 |

Since Spec-Gaussian employs a neural network for real-time rendering, it significantly impacts the FPS. This also affects scene reconstruction speed. Our method requires less memory and achieves a much higher FPS. However, due to our two-stage strategy (first training, then applying post-processing), the initial 3DGS optimization accounts for ~50% of the overall training time. Consequently, the total training time on a single GPU is comparable to that of Spec-Gaussian. Using multiple GPUs could significantly reduce the duration of the 2D Gaussian optimization and back-projection stages.

[1] Spec-Gaussian: Anisotropic View-Dependent Appearance for 3D Gaussian Splatting

[2] AbsGS: Recovering Fine Details for 3D Gaussian Splatting

---

> ### Comment · Reviewer_mqPT · 2025-11-26
> **Concerns regarding evaluations of Spec-Gaussian**
>
> Thank you for the clarification. However, I remain concerned about the evaluation of Spec-Gaussian [A]. Independent works, including Normal-GS [B] and SpecGaussian with Latent Features [C], report significantly higher performance for Spec-Gaussian than what is shown in your response, despite [B] using only a subset of scenes in MipNerf360 due to licensing limitations. To better understand this discrepancy, could you report per-scene metrics for Spec-Gaussian and explain the source of the performance gap? Moreover, the original SpecGaussian [A] relies on the 3DGS representation rather than the 2DGS representation.
>
> From [B]:
> Spec-Gaussian achieves (PSNR, SSIM, LPIPS) values of (24.50, 0.855, 0.175) on Tanks and Temples (2 scenes) and (30.114, 0.905, 0.252) on Deep Blending (2 scenes).
>
> From [C]:
> Spec-Gaussian achieves (PSNR, SSIM, LPIPS) values of (24.46, 0.864, 0.160) on Tanks and Temples (2 scenes) and (30.41, 0.912, 0.240) on Deep Blending (2 scenes).
>
> [A] Spec-Gaussian: Anisotropic View-Dependent Appearance for 3D Gaussian Splatting. NeuralIPS 2024
>
> [B] Normal-GS: 3D Gaussian Splatting with Normal-Involved Rendering. NeuralIPS 2024
>
> [C] SpecGaussian with Latent Features: A High-quality Modeling of the View-dependent Appearance for 3D Gaussian Splatting. ACM MM. 2024

---

> ### Author Response · Authors · 2025-11-26
> **Response to Reviewer mqPT**
>
> We are pleased to receive your response. First, we would like to assure you that all our data are the results of genuine reproduction. Regarding the discrepancy in metrics (which appears to be a misunderstanding), we have provided an explanation in Section 2.
> ## 1. Per-scene Metrics
>
> The Spec-Gaussian\[A\] paper provides per-scene metrics, which we have compiled in the table below:
>
> |        | bicycle | flowers | garden | stump | treehill | room  | counter | kitchen | bonsai | Mean  |
> | ------ | ------- | ------- | ------ | ----- | -------- | ----- | ------- | ------- | ------ | ----- |
> | PSNR↑  | 25.90   | 21.86   | 28.07  | 27.25 | 22.48    | 32.11 | 30.12   | 32.25   | 33.54  | 28.18 |
> | SSIM↑  | 0.797   | 0.648   | 0.881  | 0.797 | 0.647    | 0.935 | 0.923   | 0.937   | 0.953  | 0.835 |
> | LPIPS↓ | 0.166   | 0.263   | 0.092  | 0.184 | 0.269    | 0.177 | 0.166   | 0.108   | 0.162  | 0.176 |
>
> The following are our reproduced results for Spec-Gaussian:
>
> |        | bicycle | flowers | garden | stump | treehill | room  | counter | kitchen | bonsai | Mean  |
> | ------ | ------- | ------- | ------ | ----- | -------- | ----- | ------- | ------- | ------ | ----- |
> | PSNR↑  | 25.80   | 21.64   | 27.94  | 27.03 | 22.24    | 31.73 | 30.01   | 32.10   | 33.27  | 27.97 |
> | SSIM↑  | 0.797   | 0.648   | 0.880  | 0.794 | 0.645    | 0.933 | 0.923   | 0.936   | 0.952  | 0.834 |
> | LPIPS↓ | 0.167   | 0.264   | 0.093  | 0.186 | 0.270    | 0.176 | 0.165   | 0.107   | 0.160  | 0.176 |
>
> For the complete per-scene metrics of our method, please refer to Tables 6, 7, and 8 in the supplementary material. We also list here the per-scene metrics of our method using 3rd-order Spherical Harmonics within the MCMC framework:
>
> |        | bicycle | flowers | garden | stump | treehill | room  | counter | kitchen | bonsai | Mean  |
> | ------ | ------- | ------- | ------ | ----- | -------- | ----- | ------- | ------- | ------ | ----- |
> | PSNR↑  | 26.32   | 22.44   | 28.36  | 27.90 | 23.46    | 33.45 | 30.82   | 33.15   | 34.78  | 28.96 |
> | SSIM↑  | 0.814   | 0.663   | 0.887  | 0.825 | 0.680    | 0.942 | 0.930   | 0.941   | 0.958  | 0.849 |
> | LPIPS↓ | 0.158   | 0.270   | 0.086  | 0.161 | 0.259    | 0.170 | 0.158   | 0.106   | 0.159  | 0.170 |
>
> ## 2. Cause of Performance Gap
>
> The metrics provided by SpecGaussian\[C\] are from the version **trained  with anchors** (as introduced in \[D\]) provided by Spec-Gaussian (NormalGS\[B\] might also use this version, although it is not specified in their paper). According to the description by the Spec-Gaussian authors, generally better rendering quality can be achieved without using anchors. Therefore, the metrics we presented in our rebuttal originate from the version without anchors.
>
> However, to our surprise, on the Tanks and Temples and Deep Blending datasets, we found that  using anchors actually leads to better rendering quality. We present the reproduced results in the table below:
>
> |   |       | T&T   |        | \|  | Deep  | Blending |        |
> | -------------------------- | ----- | ----- | ------ | --- | ----- | -------- | ------ |
> |   | PSNR↑ | SSIM↑ | LPIPS↓ |     | PSNR↑ | SSIM↑    | LPIPS↓ |
> | Spec-Gaussian (w/o anchor) | 23.77 | 0.855 | 0.166  |     | 29.53 | 0.905    | 0.241  |
> | Spec-Gaussian (w/ anchor)  | 24.37 | 0.856 | 0.173  |     | 30.26 | 0.908    | 0.247  |
>
> Thus, using anchors in Spec-Gaussian improves rendering quality on the aforementioned two datasets but degrades it on the Mip-NeRF 360 dataset. **Overall, its performance still cannot surpass our best metrics**. Furthermore, we believe that comparisons with anchor-based methods lack justification. This method was initially proposed by \[D\], which uses an MLP to generate neural Gaussians around SFM-initialized anchor Gaussians. While it saves GPU memory and enhances view-dependent effects, the cost is the loss of the explicit Gaussian representation. Another inconvenience of anchor-based methods is the need to specify the hyperparameter of voxel size. In Spec-Gaussian, they set different voxel sizes (0.001, 0.01, 0.005) for Mip-NeRF 360, T&T, and Deep Blending respectively to achieve optimal results. **In contrast, our method does not require tuning separate hyperparameters.**
>
> We believe that comparison with the anchor-free variant of Spec-Gaussian is more practically meaningful. Against this version, our method achieves twice the rendering speed without introducing additional time or space overhead.
>
> Thank you for your response. If you have further questions, please feel free to ask, and we will address them promptly.
>
> \[A\] Spec-Gaussian: Anisotropic View-Dependent Appearance for 3D Gaussian Splatting. NeuralIPS 2024
>
> \[B\] Normal-GS: 3D Gaussian Splatting with Normal-Involved Rendering. NeuralIPS 2024
>
> \[C\] SpecGaussian with Latent Features: A High-quality Modeling of the View-dependent Appearance for 3D Gaussian Splatting. ACM MM. 2024
>
> \[D\] Scaffold-gs: Structured 3d gaussians for view-adaptive rendering. CVPR. 2024.

---

### Author Response · Authors · 2025-11-17
**Response to Common issues (part 2)**

# Response to Common issues (part 2)
## 2. Using different primitives
We note that reviewers hold different attitudes regarding **whether different primitives should be used**. We fully understand this disagreement. While our approach might be less conducive to downstream tasks such as inverse rendering and relighting, we have demonstrated its advantages in **improving rendering quality**. Let's draw an illustrative example from modern CG and game rendering for discussion—we often design different shaders for different material types, for two reasons: 1) No unified framework exists that can model all types of materials; 2) The computational complexity of shading differs across materials. (e.g., the Disney BSDF shading model involves far more complex calculations than simple diffuse reflection.) Using simpler shaders effectively saves computation time. This leads us to a key insight—If using different shaders is common practice in traditional triangle-based rendering pipelines, then why shouldn't Gaussian splatting employ different primitives? Given the vast diversity of materials in nature, attempting to model them all within a unified framework is likely infeasible (though neural networks could attempt this at the cost of heavy rendering overhead, as seen in Spec-Gaussian). Our solution, which introduces a second explicit primitive, offers the benefit of significantly improving rendering quality with minimal computational overhead.

In the paper, we chose DBS as the main baseline because this work utilizes Phong shading to model specular highlights and, at the time of submission, it had set the state-of-the-art in rendering quality among splatting-based methods. The goal of our work is to **push the rendering quality of 3D Gaussians to its limits**. We argue that **using different primitives represents a promising direction for this challenging problem**. Our work takes the first step along this path, and we believe the rendering quality of 3D Gaussians is far from reaching its full potential. Ample future work remains. For instance, one could explore using different primitives to model different materials from the very beginning of training, rather than introducing new primitives only at the post-processing stage. Another direction is designing a wider variety of primitives with adaptive assignment to different surfaces based on complexity needs.

---

### Author Response · Authors · 2025-11-17
**Supplementary comparison**

Dear reviewers, we have just noticed a concurrent paper[1] submitted to ICLR 2026 that received very positive reviews (with scores 6, 8, 4, 6). This work proposes N-dimensional anisotropic Beta kernels for explicit radiance field rendering. Interestingly, compared to their method, we achieved a lead on the real-world dataset Mip-NeRF 360, but slightly lagged behind on NeRF Synthetic. We believe our method will also achieve a lead on other real-world datasets, such as T&T and Deep Blending. Nevertheless, we consider the greatest strength of our method lies in its simplicity and plug-and-play versatility.

| Method           |        Mip-         |        NeRF         |         360         | \|  |                     |         T&T         |                     | \|  |        Deep         |      Blending       |                     | \|  |        NeRF         |      Synthetic      |                     |
| ---------------- | :-----------------: | :-----------------: | :-----------------: | :-: | :-----------------: | :-----------------: | :-----------------: | :-: | :-----------------: | :-----------------: | :-----------------: | :-: | :-----------------: | :-----------------: | :-----------------: |
|                  |        PSNR↑        |        SSIM↑        |       LPIPS↓        |     |        PSNR↑        |        SSIM↑        |       LPIPS↓        |     |        PSNR↑        |        SSIM↑        |       LPIPS↓        |     |        PSNR↑        |        SSIM↑        |       LPIPS↓        |
| UBS-6D [1]       |        28.66        |        0.840        |        0.184        |     |          -          |          -          |          -          |     |          -          |          -          |          -          |     |  $\textbf{34.92}$   |  $\textbf{0.975}$   |  $\textbf{0.026}$   |
| Ours (3DGS,sh=3) |        28.39        |        0.834        |        0.184        |     |        23.90        |        0.856        |        0.158        |     |        29.90        | $\underline{0.903}$ |        0.237        |     |        34.02        |        0.969        |        0.031        |
| Ours (MCMC,sh=2) | $\underline{28.89}$ | $\underline{0.848}$ | $\underline{0.171}$ |     | $\underline{25.04}$ | $\underline{0.871}$ | $\underline{0.146}$ |     |  $\textbf{30.33}$   |  $\textbf{0.909}$   |  $\textbf{0.235}$   |     |        34.03        |        0.970        |        0.030        |
| Ours (MCMC,sh=3) |  $\textbf{28.96}$   |  $\textbf{0.849}$   |  $\textbf{0.170}$   |     |  $\textbf{25.06}$   |  $\textbf{0.872}$   |  $\textbf{0.144}$   |     | $\underline{30.22}$ |  $\textbf{0.909}$   | $\underline{0.236}$ |     | $\underline{34.35}$ | $\underline{0.971}$ | $\underline{0.029}$ |


[1] Universal Beta Splatting. [ICLR submission 7043](https://openreview.net/forum?id=51JEkjP0gF).

---

### Author Response · Authors · 2025-11-24
**Asking for additional feedback**

Dear AC and Reviewers,

We are deeply grateful to the reviewers for their thoughtful comments, which have helped us to improve our work. In our previous response, we carefully incorporated all suggestions, including **conducting the requested comparisons** (to Spec-Gaussian) and **clarifying our contributions** (pushing the rendering quality of 3DGS to its limit). We have done our best to address all points raised and hope that our revisions and responses meet the reviewers' expectations. We would be honored if the reviewers could reassess our manuscript and welcome any additional feedback they might have.

Best,

The authors

---

### Meta-Review · Area_Chair_4T1h · 2025-12-29

**Summary:**

# Decision

This submission presents a technically sound and well-motivated approach for improving 3DGS reconstruction through a post-hoc residual correction and Phong-inspired representation for specular effects. Experiments demonstrate consistent improvements over SOTA baselines in both qualitative and quantitative metrics, and the authors have addressed some key reviewer concerns with additional baselines, ablations, and efficiency measurements.

However, some limitations remain. In particular, interpretability and material-level modeling may be compromised by the post-hoc design, and the influence of certain design choices (e.g., residual-specular separation, lobe parameters) on downstream utility might remain ambiguous. Despite these caveats, the technical novelty and convincing empirical gains justify acceptance.

------------
# Consolidated Reviews

## Strengths

### Technical contributions
- Technical contribution: correcting problematic regions through two-stage 2D residual fixing, inverse splatting, and joint optimization ;  [`zmRh`, `mqPT`, `eJHV`]
- Technical contribution: representation inspired by Phong modeling to effectively model specular reflections and high-frequency lighting effects [`LiBR`, `eJHV`]
- Solution serving as a post-hoc module on top of the 3DGS-based methods [`mqPT`, `LiBR`]
- Solution addressing a meaningful problem (under-reconstruction of 3DGS) [`zmRh`]

### Convincing evaluation
- Convincing results outperforming SOTA baselines (qualitative and quantitative) [`zmRh`,  `eJHV`]

## Weaknesses

### Validation missing key baselines and discussions
- Lack of comparison with relevant baselines (e.g., SpecGaussian) [`zmRh`, `mqPT`]
- No evaluation w.r.t. impact of proposed scheme on depth accuracy (c.f. use of geometric information for back-projection) [`LiBR`]
- Lack of evaluation w.r.t. training stability or convergence behavior [`eJHV`]
- Lack of analysis w.r.t. influence of lobe parameters on results [`eJHV`]

### Lack of insight regarding design's downsides
- Unclear positioning w.r.t. SH representation [`mqPT`, `LiBR`]
- Possibly sub-optimal separation of diffuse and specular components across different primitives, tying specular capacity to residual patterns rather than material properties, and risks uneven coverage and missed interactions, sacrificing some interpretability and downstream utility [`mqPT`]
- Downsides of post-hoc module: risk of accumulating biases/errors from earlier stages, extra computation and memory costs [`eJHV`]

**Reviewer Concerns:**

See above for summary of main concerns shared by reviewers.

Overall, the authors have thoroughly tackled the reviewers' concerns, providing new results covering additional SOTA baselines (SpecGaussian, AbsGS, UBS) and metrics (FPS, memory). They have also shared some insights regarding their design choices, with a _the-end-justifies-the-mean_ approach to improving novel-view-synthesis accuracy. Still, some concerns regarding the loss of interpretability and modeling capability might remain valid.

**Reviewer Scores:**

### Reviewer `zmRh`
- **Original score:** 6
- **Score change:** increased to 8, c.f. the reviewer's own reply.

### Reviewer `mqPT`
- **Original score:** 2
- **Score change:** likely to have increased, e.g., to 4 or 6, c.f. request for authors to provide additional results in exchange for a score increase (which the authors did).

### Reviewer `LiBR`
- **Original score:** 4
- **Score change:** likely to have kept their score, c.f. unresponsive + some of their concerns might still be considered outstanding.

### Reviewer `eJHV`
- **Original score:** 4
- **Score change:** likely to have kept their score, c.f. unresponsive ; but might have increased too, c.f. concerns covered by authors.

---

### Decision · Program_Chairs · 2026-01-26

Accept (Poster)